# Online Bilateral Trade With Minimal Feedback: Don't Waste Seller's Time

**Francesco Bacchiocchi**
Politecnico di Milano
francesco.bacchiocchi@polimi.it

**Matteo Castiglioni**
Politecnico di Milano
matteo.castiglioni@polimi.it

**Roberto Colomboni**
Politecnico di Milano &
Università degli Studi di Milano
roberto.colomboni@polimi.it

**Alberto Marchesi**
Politecnico di Milano
alberto.marchesi@polimi.it

## Abstract

Online learning algorithms for designing optimal bilateral trade mechanisms have recently received significant attention. This paper addresses a key inefficiency in prior two-bit feedback models, which synchronously query both the buyer and the seller for their willingness to trade. This approach is inherently inefficient as it offers a trade to the seller even when the buyer rejects the offer. We propose an *asynchronous* mechanism that queries the seller only if the buyer has already accepted the offer. Consequently, the mechanism receives one bit of feedback from the buyer and a "censored" bit from the seller—a signal richer than the standard one-bit (trade/no-trade) feedback, but less informative than the two-bit model. Assuming independent valuations with bounded densities—the same distributional conditions underlying the two-bit results of Cesa-Bianchi et al. [2024a]—we design an algorithm that achieves $\tilde{O}(T^{2/3})$ regret against the best fixed price in hindsight. This matches the lower bound for the strictly richer two-bit model, showing that our mechanism elicits the minimal feedback necessary to attain optimal rates.

## 1 Introduction

In a bilateral trade problem, a broker faces two rational agents—a *seller* and a *buyer*—who wish to trade an object. Each agent has their own private valuation for the object and seeks to maximize their utility. The goal of the broker is to design a mechanism that intermediates between the seller and the buyer, in order to make a trade happen. Ideally, a mechanism for bilateral trade should be *efficient*, *i.e.*, it should maximize the sum of agents' utilities, while also ensuring incentive compatibility and individual rationality. A well-known mechanism that achieves this objective is the VCG mechanism [Vickrey, 1961]. Unfortunately, the VCG mechanism fails to meet *budget balance*, requiring the broker to subsidize the market and incur in financial losses. Indeed, a general impossibility result by Myerson and Satterthwaite [1983] shows that full efficiency cannot be attained while simultaneously maintaining incentive compatibility, individual rationality, and budget balance.

A recent line of research (see, *e.g.*, [Cesa-Bianchi et al., 2021, 2024a, Azar et al., 2022]) circumvents the impossibility result by Myerson and Satterthwaite [1983] through the lens of online learning. This is done by addressing repeated bilateral trade problems, where the broker faces a sequence of sellers and buyers willing to trade objects, over a time horizon $T$. At each time $t$, a new seller and a new buyer arrive, each of them with their own private valuation of the object, say $S_t$ and $B_t$, respectively. The broker then proposes a trading price $P_t$ to both agents. Thus, the seller is willing to trade if $S_t \leq P_t$, while the buyer if $B_t \geq P_t$. The trade happens only if both agents accept the price $P_t$, with

39th Conference on Neural Information Processing Systems (NeurIPS 2025).

the buyer paying the trading price to the seller and receiving the object, resulting in *strong budget balance* (*i.e.*, the broker neither subsidize nor extract revenue from the market). In such an online learning framework, the full-efficiency requirement is relaxed by comparing the performance of the broker over the $T$ time steps against the *best fixed price in hindsight*. Specifically, the performance is evaluated in terms of the *gain from trade*, which intuitively encodes the net gain in agents' utilities, defined as $(B_t - S_t)\mathbb{I}\{S_t \le P_t \le B_t\}$ at each time $t$.

Previous works have focused on three models that differ in the kind of feedback that the broker receives at the end of each time step $t$. In the *full-feedback* model, the broker observes the valuations $S_t$ and $B_t$ of the agents. In the *two-bit* model, the broker separately observes whether each of the two agents is willing to trade or not, namely they observe both $\mathbb{I}\{S_t \le P_t\}$ and $\mathbb{I}\{P_t \le B_t\}$. Finally, in the *one-bit* model, they only observe whether the trade has occurred or not, namely $\mathbb{I}\{S_t \le P_t \le B_t\}$.

While simple, the one-bit model is insufficient for learning optimal strongly-budget-balanced mechanisms [Cesa-Bianchi et al., 2024a]. This motivates the study of richer feedback models, such as the two-bit one, where learning becomes possible under the assumption of independent seller/buyer valuations with bounded densities [Cesa-Bianchi et al., 2024a]. However, such mechanisms rely on synchronous interaction protocols that are inherently inefficient: they query the seller even when the buyer has already rejected the trade. In this paper, we address the following two natural questions:

*What is the minimal feedback required to learn optimal mechanisms?*
*Is it possible to query the seller only when a trade opportunity arises?*

From a more application-oriented perspective, these questions raise from the insight that it would generally be more efficient—and more reasonable—to implement *asynchronous* interaction protocols, in which the seller is approached only if the buyer has already agreed to trade at the proposed price. This approach is especially relevant in many practical applications—such as, *e.g.*, online freelance marketplaces (Upwork, Fiverr), ride-sharing platforms (Uber, Lyft, Grab), rental intermediaries platforms (AirBnB)—where sellers are often involved in multiple simultaneous trading scenarios involving different objects. In such settings, requiring the sellers to make a decision each time a potential buyer for any of their objects appears would impose an excessive burden on them. Furthermore, in many cases, sellers prefer to disclose as little information as possible to the broker in order to protect their reputation.

**Our Results**  We study—for the first time, to the best of our knowledge—*online learning in bilateral trade problems with asynchronous interaction protocols*. The key challenge in such a setting is that the broker receives a particular *asymmetric* feedback that is richer than one-bit feedback, but way less informative than two-bit feedback. Specifically, the broker receives one bit of feedback from the buyer, by observing $\mathbb{I}\{B_t \ge P_t\}$ at every time $t$, while they only observe a "censored" bit of feedback from the seller, as they get to know $\mathbb{I}\{S_t \le P_t\}$ only when the buyer accepts the trade. The main result of the paper is a strongly-budget-balanced algorithm that attains $\tilde{O}(T^{2/3})$ regret against the best fixed price in hindsight, assuming independent sellers and buyers' valuations with bounded densities. Notice that both assumptions are required, since removing even one of the two assumptions makes the problem not learnable even with two-bit feedback [Cesa-Bianchi et al., 2024a]. We remark that our result matches the regret rate that Cesa-Bianchi et al. [2024a] obtained using the more informative two-bit feedback, under the *same distributional assumptions*. Moreover, it matches the lower bound by Cesa-Bianchi et al. [2024a] for the richer two-bit feedback model, showing that:

*Asynchronous protocols are not only efficient, but they also allow the broker to elicit the minimal feedback necessary to attain optimal regret rates!*

## 1.1   Challenges and Techniques

Our algorithm builds upon the idea of *scouting bandits*, which have been originally introduced by Cesa-Bianchi et al. [2024a] for two-bit models. This idea exploits a suitable decomposition of the *expected* gain from trade $g(p)$ for a fixed price $p \in [0, 1]$, which is defined as follows:

$$g(p) = \underbrace{\mathbb{P}[S \le p]}_{(a)} \underbrace{\int_p^1 \mathbb{P}[B \ge \lambda]\,\mathrm{d}\lambda}_{(b)} + \underbrace{\mathbb{P}[B \ge p]}_{(c)} \underbrace{\int_0^p \mathbb{P}[S \le \lambda]\,\mathrm{d}\lambda}_{(d)},$$

where $S$ and $B$ are generic random variables representing the valuations of the seller and the buyer, respectively. Cesa-Bianchi et al. [2024a] leverage the two-bit feedback by first conducting a uniform exploration phase to estimate the integral terms (b) and (d). Using these estimates, they construct a proxy for the expected reward function in which (b) and (d) are replaced with their empirical counterparts. Then, they run a *bandit* algorithm with this proxy, since the two-bit feedback provides, at each time step, unbiased estimates of terms (a) and (c), thereby allowing the learner to reconstruct the proxy reward function.

Our *asymmetric feedback model introduces a new significant challenge*: the feedback received from the seller is "censored", as the learner gets to know the value of $\mathbb{I}\{S_t \leq P_t\}$ only when the buyer accepts the trade, *i.e.*, when $\mathbb{I}\{P_t \leq B_t\} = 1$. This "censored" seller's feedback makes the scouting bandits approach by Cesa-Bianchi et al. [2024a] unsuitable for our setting: the estimator they build for the integral term (d) cannot be recovered due to missing observations, and the bandit feedback needed to estimate term (a) may be "censored" and thus unavailable. Furthermore, our feedback breaks the symmetry in estimating the four components in the decomposition of the expected gain from trade provided by the two-bit feedback. Indeed, in our model, estimating (a) is harder than estimating (c), and estimating (d) is harder than estimating (b). Consequently, *the main challenge faced in this paper is how to effectively estimate (a) and (d) under "censored" seller's feedback*. Our key technical contribution is addressing this challenge.[1]

In order to address the challenge, we provide a lower bound on the number of time steps in which the value of $\mathbb{I}\{S_t \leq p\}$ (*i.e.*, the seller's feedback for $p$) is observed, for every price $p$. In particular, if price $p$ is posted for $H$ times (and $\mathbb{P}[B \geq p]$ is large enough), we can lower bound the number of times that seller's feedback is observed as $\Omega\big(H \cdot \mathbb{P}[B \geq p]\big)$, by using Chernoff's concentration bound. Notably, the usual additive concentration bounds are of no help in this setting. Indeed, our random variables are Bernoulli that might have small mean, making additive bounds non-meaningful. Moreover, the lower bound on the number of times the seller's feedback is observed could still be small. As a consequence, we may lack sufficient samples to build precise confidence bounds around terms (a) and (d), thus precluding the direct application of standard UCB-like techniques. However, we observe that when the confidence intervals on seller-related quantities are large, the buyer's probability of accepting the trade is low, resulting in two effects that counterbalance each other. More formally, the error on term (d) is scaled by the buyer's acceptance probability—namely, term (c). A symmetric argument holds for terms (a) and (b).

## 1.2 Related Works

Our work contributes to the line of research initiated by Cesa-Bianchi et al. [2024a], which studies bilateral trade through the lens of online learning. Among other results, Cesa-Bianchi et al. [2024a] show that strongly-budget-balanced mechanisms are learnable with two-bit feedback when the seller/buyer distributions are independent and admit bounded density, while the same problem is unlearnable under one-bit model.

Subsequent work focuses mainly on adversarial settings. Azar et al. [2022] design algorithms that guarantee no-2-regret. Cesa-Bianchi et al. [2024b] provide sublinear regret guarantees assuming a smoothed adversary. Bernasconi et al. [2024], Chen et al. [2025] remove the smoothness assumption by relaxing the budget balance constraint to hold globally. Lunghi et al. [2026] go even further by analyzing which regret rates are attainable by relaxing the global budget constraint (*i.e.*, by allowing for its violation). Other works study extensions of bilateral trade to multiple buyers [Babaioff et al., 2024, Lunghi et al., 2025], different objectives [Bachoc et al., 2024], contextual settings [Gaucher et al., 2025], divisible items [Bolić et al., 2025], and situations where traders have no predetermined seller and buyer's roles [Bolić et al., 2024, Cesari and Colomboni, 2025, Bachoc et al., 2025a,b].

There is also a rich literature on bilateral trade without learning, focusing on providing approximations of an optimal mechanism [Colini-Baldeschi et al., 2016, 2017, Blumrosen and Mizrahi, 2016, Brustle et al., 2017, Colini-Baldeschi et al., 2020, Babaioff et al., 2020, Dütting et al., 2021, Deng et al., 2022, Kang and Vondrák, 2019, Archbold et al., 2023].

---

[1]We remark that, from a technical point of view, the roles of the seller and the buyer in our framework are completely interchangeable without requiring additional effort. Indeed, the case in which we query the buyer only when the seller agrees to trade at a given price can be tackled with the same approach presented here.

## 2 Preliminaries

In this paper, we study online learning in repeated bilateral trade problems. In this section, we introduce the notation and all the definitions needed in the rest of the paper.

### 2.1 Bilateral Trade

The learner (a broker) repeatedly interacts with the environment. At each time step $t \in [T]$, a new seller and a new buyer arrive with (random) valuation $S_t$ and $B_t$ in $[0, 1]$.[2]

The learner offers a (random) price $P_t \in [0, 1]$ to the buyer and the seller. A trade happens if and only if the buyer and the seller accept the proposed price, *i.e.*, when $S_t \le P_t \le B_t$. This ensures *strong budget balance* is satisfied during learning. The learner's performance is evaluated through the net increase in market value (*i.e.*, the net increase in agents' utilities), also known as *gain from trade*. Specifically, if we define the gain from trade function as

$$\text{gft} \colon [0,1]^3 \to [0,1], \qquad (p, s, b) \mapsto (\underbrace{b - p}_{\substack{\text{buyer's} \\ \text{net gain}}} + \underbrace{p - s}_{\substack{\text{sellers's} \\ \text{net gain}}}) \underbrace{\mathbb{I}\left\{s \le p \le b\right\}}_{\text{a trade happens}} = (b - s)\mathbb{I}\left\{s \le p \le b\right\}$$

and the gain from trade (random) function at time $t$ as

$$\text{GFT}_t \colon [0, 1] \to [0, 1], p \mapsto \text{gft}(p, S_t, B_t),$$

the gain from trade rewarded to the learner by posting $P_t$ is the random variable $\text{GFT}_t(P_t)$.

We assume that the sequence of sellers' valuations $S_1, S_2, \ldots$ and the sequence of buyers' valuations $B_1, B_2, \ldots$ are i.i.d. sequences, independent of each other. Moreover, for ease of presentation, we introduce two additional random variables $S$ and $B$, which are distributed as $S_t$ and $B_t$, respectively. We assume that $S$ and $B$ are independent of each other and of the two sequences $S_1, S_2, \ldots$ and $B_1, B_2, \ldots$, and that they admit $L$-Lipschitz continuous cumulative distribution functions (cdf)[3].

For notational convenience, we also define the random function

$$\text{GFT} \colon [0, 1] \to [0, 1], \qquad p \mapsto \text{gft}(p, S, B) \,,$$

and the expected gain from trade function as

$$\text{g} \colon [0, 1] \to [0, 1], \qquad p \mapsto \mathbb{E}[\text{GFT}(p)] \,.$$

We notice that [Cesa-Bianchi et al., 2024a, Lemma 2] ensures that g is upper semicontinuous and, consequently, being defined on the compact set $[0, 1]$, it admits a maximum. From this point on, we fix a point $p^\star \in [0, 1]$ where this maximum is attained.

### 2.2 Do Not Waste the Seller's Time: An Asynchronous Protocol

It is well known (see [Cesa-Bianchi et al., 2024a]) that learning is impossible with one-bit feedback, *i.e.*, when the learner only observes the outcome of the trade $\mathbb{I}\{S_t \le P_t \le B_t\}$ after each time step $t$. For this reason, previous works focus on two-bit feedback, where the leaner can separately observe seller and buyer's willingness to trade, namely $\mathbb{I}\{S_t \le P_t\}$ and $\mathbb{I}\{P_t \le B_t\}$. In this paper, we show that the regret attainable with two-bit feedback can also be obtained with a weaker feedback, under the same assumptions. In particular, we consider an *asynchronous* protocol that first proposes a trading price to the buyer, and, then, it offers the same price to the seller only if the buyer has already expressed their willingness to trade at the proposed price. This protocol introduces a new *asymmetric* feedback model in which the leaner receives one bit of feedback from the buyer, while it gets a "censored" bit from the seller, as their willingness to trade is observed only when the buyer accepts to the trade at the proposed price.

The asynchronous interaction protocol between the learner (a broker) and the environment (the sequence of sellers and buyers) is formally presented in Online Protocol 1.

---

[2] Let $n \in \mathbb{N}$, we denote with $[n] = \{1, \ldots, n\}$ the set of the first $n$ natural numbers.

[3] The cdf Lipschitz assumption is equivalent to the bounded density one of Cesa-Bianchi et al. [2024a]

---

**Online Protocol 1** Asynchronous Repeated Bilateral Trade

---
1: **for** time step $t = 1, 2 \ldots$ **do**
2:     The learner chooses $P_t \in [0, 1]$
3:     The learner observes $\mathbb{I}\{P_t \leq B_t\}$
4:     If $P_t \leq B_t$, then the learner observes $\mathbb{I}\{S_t \leq P_t\}$
5:     The learner gains (but does *not* observe) $\mathrm{GFT}_t(P_t)$

---

**Regret**    Given a time horizon $T \in \mathbb{N}$, the goal of the learner is to minimize the regret with respect to the gain from trade, defined as

$$R_T = \sum_{t=1}^{T} \Big( \mathrm{g}(p^\star) - \mathbb{E}\big[\mathrm{g}(P_t)\big] \Big),$$

where $p^\star \in \arg\max_{p \in [0,1]} \mathrm{g}(p)$.

## 3 Algorithm

In this section, we present an online learning algorithm that achieves optimal regret guarantees under the asynchronous interaction protocol introduced in Section 2. Our learning algorithm is restricted to using a finite grid of prices. In Section 4, we show that this results in a small loss that depends on the granularity of the grid and the Lipschitz constant $L$. Specifically, we denote by $\mathcal{P}_K \subseteq [0, 1]$ the uniform grid of prices $p_k := {}^{k-1}/K$ for $k \in [K]$, where $K$ is a suitable parameter. More formally, we let $\mathcal{P}_K := \{p_k\}_{k \in [K]}$.

### 3.1 Additional Notation

Before moving to the description of our algorithm, we need to introduce some additional notation that is useful to deal with the stochastic feedback observed by posting prices on the grid. For each $k \in [K]$ and each $j \in \mathbb{N}$, we denote with $t_B(k, j)$ the $j$-th time step in which the broker sets price $p_k \in \mathcal{P}_K$ (if this time step exists, otherwise, we set it to $+\infty$), and with $t_S(k, j)$ the $j$-th time the broker sets price $p_k \in \mathcal{P}_K$ and the buyer accepts to buy (it this time step exists, otherwise we set it to $+\infty$). For technical reasons (*i.e.*, having well-defined random variables for every $k \in [K]$ and $j \in \mathbb{N}$), we also assume given another i.i.d. family of pairs of random variables $(S'_{k,j}, B'_{k,j})_{k \in [K], j \in \mathbb{N}}$, independent of $(S_1, B_1), (S_2, B_2), \ldots$ such that, for any $k \in [K]$ and $j \in \mathbb{N}$, the pair $(S'_{k,j}, B'_{k,j})$ shares the same distribution as $(S, B)$. For each $k \in [K]$ and $j \in \mathbb{N}$, we then set

$$B_{k,j} := \begin{cases} B_{t_B(k,j)} & \text{if } t_B(k,j) < +\infty , \\ B'_{k,j} & \text{otherwise} , \end{cases} \qquad S_{k,j} := \begin{cases} S_{t_S(k,j)} & \text{if } t_S(k,j) < +\infty , \\ S'_{k,j} & \text{otherwise} . \end{cases}$$

In this way, the family $(S_{k,j}, B_{k,j})_{k \in [K], j \in \mathbb{N}}$ is a well-defined independent family of pairs of random variables such that, for every $k \in [K]$, the sequence $(S_{k,j}, B_{k,j})_{j \in \mathbb{N}}$ is i.i.d., and, additionally, for every $j \in \mathbb{N}$, the pair $(S_{k,j}, B_{k,j})$ shares the same distribution as $(S, B)$.

### 3.2 Algorithm Description

We are now ready to introduce our algorithm achieving optimal regret guarantees under the asynchronous interacton protocol (Algorithm 2). The algorithm takes as input a time horizon $T$, and it builds the uniform grid $\mathcal{P}_K \subseteq [0, 1]$ with $K := \lceil T^{1/3} \rceil$ points. At a very high level, our algorithm performs an initial exploration phase in which the broker plays each of the prices in $\mathcal{P}_K$ a number of times equal to $H := \lceil T^{1/3} \rceil$. This exploration phase is necessary to compute an optimistic estimate of the expected gain from trade, as defined in eq. (1) in the following. Once this exploration phase is concluded, in the remaining time steps, Algorithm 2 executes a $K$-armed bandit-style algorithm by selecting, at each time, the price $p_k \in \mathcal{P}_K$ that maximizes a suitable optimistic estimate of the expected gain from trade. Notice that several additional components discussed in the following are needed to deal with the limited feedback.

---

**Algorithm 2** More than one-bit less than two-bit bilateral trade

---

1: **Input**: time horizon $T$
2: Set $\delta \leftarrow \frac{1}{T^2}, H \leftarrow K \leftarrow \lceil T^{1/3} \rceil, \mathcal{T}_{k,0} \leftarrow 0, \mathcal{Q}_{k,0} \leftarrow 0, \mathcal{S}_{k,0} \leftarrow 0 \ \forall k \in [K]$
3: Set $\mathcal{P}_K \leftarrow \{p_k\}_{k \in [K]}$ with $p_k \leftarrow {}^{k-1}/_K \ \forall k \in [K], K^\diamond \leftarrow \emptyset$
4: **for** $t = 1, 2, \ldots, T$ **do**
5:    **if** $t \leq HK$ **then**                                                       ▷ Exploration Phase
6:       Set $l \leftarrow \lceil t/H \rceil$ and post price $P_t \leftarrow p_l$
7:    **else**                                                             ▷ Bandit Phase
8:       Select $l \in \text{argmax}_{k \in K^\diamond} UCB_{k,t-1}$ (see eq. (3)) and post price $P_t \leftarrow p_l$
9:    **for** $k = 1, 2 \ldots K$ **do**                                              ▷ Update Counters
10:       **if** $t = HK$ **then**
11:          $N_k \leftarrow \min_{i \leq k} \mathcal{Q}_{i,KH}$
12:          $K^\diamond \leftarrow \{k \in [K] \mid \mathcal{Q}_{k,KH} \geq 32 \log(KT^2/\delta)\}$
13:          $\hat{F}_k \leftarrow \frac{1}{KH} \sum_{i=k}^{K-1} \sum_{j=1}^{H} \mathbb{I}\{B_{i,j} \geq p_i\}, \ \forall k \in K^\diamond$
14:          $\hat{G}_k \leftarrow \frac{1}{KN_k} \sum_{i=1}^{k} \sum_{j=1}^{N_k} \mathbb{I}\{S_{i,j} \leq p_i\}, \ \forall k \in K^\diamond$
15:       Set $\mathcal{T}_{k,t} \leftarrow \mathcal{T}_{k,t-1} + \mathbb{I}\{P_t = p_k\}$
16:       Set $\mathcal{Q}_{k,t} \leftarrow \mathcal{Q}_{k,t-1} + \mathbb{I}\{P_t = p_k\}\mathbb{I}\{P_t \leq B_t\}$
17:       Set $\mathcal{S}_{k,t} \leftarrow \mathcal{S}_{k,t-1} + \mathbb{I}\{P_t = p_k\}\mathbb{I}\{S_t \leq P_t\}\mathbb{I}\{P_t \leq B_t\}$
18:       Set $\hat{\nu}_{k,t} \leftarrow \frac{\mathcal{Q}_{k,t}}{\mathcal{T}_{k,t}}$ and $\hat{\mu}_{k,t} \leftarrow \frac{\mathcal{S}_{k,t}}{\mathcal{Q}_{k,t}}$

---

We now provide a more detailed description of how the algorithm works. Specifically, in the first $KH$ time steps, Algorithm 2 prescribes the broker to play each price on the grid $\mathcal{P}_K$ exactly $H$ times—the first $H$ time steps on $p_1$, the next $H$ on $p_2$, and so forth—so that this initial phase has length $KH$. Furthermore, independently of the round $t \in [T]$, Algorithm 2 maintains counters that track how many times each price $p_k$ has been selected. Precisely, at each round $t \in [T]$: $\mathcal{T}_{k,t}$ counts the number of times the broker has proposed price $p_k$ up to round $t \in [T]$; $\mathcal{Q}_{k,t}$ counts the number of times the buyer has accepted to buy at price $p_k$; and $\mathcal{S}_{k,t}$ counts the number of times both the buyer and the seller have agreed to trade at price $p_k$.

Once the exploration phase is completed—*i.e.*, after $KH$ rounds—Algorithm 2 identifies a subset of arms $K^\diamond \subseteq [K]$ such that, for each $k \in K^\diamond$, the number of samples collected from the seller's distribution when the broker has proposed price $p_k$ is larger than a suitable constant. After that, the algorithm ignores all the arms *not* in $K^\diamond$, while for each arm in $K^\diamond$ it designs an upper confidence bound on the gain from trade (refined at each round), by exploiting the decomposition:

$$\text{g}(p) = \underbrace{\mathbb{P}[S \leq p]}_{(a)} \underbrace{\int_p^1 \mathbb{P}[B \geq \lambda]\, d\lambda}_{(b)} + \underbrace{\mathbb{P}[B \geq p]}_{(c)} \underbrace{\int_0^p \mathbb{P}[S \leq \lambda]\, d\lambda}_{(d)}, \tag{1}$$

which is formally derived in [Cesa-Bianchi et al., 2024a, Lemma 1].

Furthermore, order to build the upper confidence bounds to be used in the subsequent bandit-style procedure, for each $k \in K^\diamond$, after the exploration phase Algorithm 2 computes two estimates of the integral terms (b) and (d) in eq. (1), which are defined as follows:

$$\hat{F}_k = \frac{1}{KH} \sum_{i=k}^{K-1} \sum_{j=1}^{H} \mathbb{I}\{B_{i,j} \geq p_i\}, \quad \hat{G}_k = \frac{1}{KN_k} \sum_{i=1}^{k} \sum_{j=1}^{N_k} \mathbb{I}\{S_{i,j} \leq p_i\}, \tag{2}$$

where

$$N_k := \min_{i \leq k} \mathcal{Q}_{i,KH}.$$

Let us remark that the definition of the term $N_k$ is necessary to ensure that, in the sum in $\hat{G}_k$, we have the same number of observations of the seller's valuation for each price $p_i$ with $i \leq k$. Indeed, while for every $p_k$ we always observe whether the buyer accepts to buy or not, we do not have analogous

information for the seller, since the observation of seller's feedback is conditioned on the buyer's decision to buy at the given price.

In the remaining rounds $t > KH$, Algorithm 2 performs a UCB-style procedure using $K^\diamond$ as the set of arms. This refinement of the arm set is necessary, as we can guarantee useful concentration properties only for the arms in $K^\diamond$. Specifically, at each round $t > KH$, Algorithm 2 computes optimistic upper confidence bounds $UCB_{k,t}$ on the value of each $g(p_k)$ for prices $p_k$ such that $k \in K^\diamond$, formally defined as follows:

$$UCB_{k,t} := \left( \hat{\mu}_{k,t} + \sqrt{\frac{\log(2T/\delta)}{2\mathcal{Q}_{k,t}}} \right) \left( \hat{F}_k + \sqrt{\frac{2\log(2/\delta)}{HK}} + \frac{1}{K} \right)$$
$$+ \left( \hat{\nu}_{k,t} + \sqrt{\frac{\log(2T/\delta)}{2\mathcal{T}_{k,t}}} \right) \left( \hat{G}_k + \sqrt{\frac{2\log(2/\delta)}{KN_k}} + \frac{1}{K} \right). \quad (3)$$

One of the main challenges in our algorithm analysis will be to show that the upper confidence bounds concentrate on the true mean. This is particularly challenging since $\mathcal{Q}_{k,t}$ might be small.

## 4 Regret Analysis

In this section, we prove the regret guarantees attained by Algorithm 2. In the following, for the ease of presentation and to avoid repetitions, we assume that a time horizon $T$ is given as a parameter, and we set every occurrence of $\delta$ to $\frac{1}{T^2}$, and every occurrence of $K$ and $H$ to $\lceil T^{1/3} \rceil$, as in Algorithm 2.

Our algorithm can be easily extended to an anytime version that does not need knowledge of $T$ by using the standard doubling trick [Cesa-Bianchi and Lugosi, 2006].

### 4.1 Restrict the Set of Candidate Prices

In the following, we will show that our algorithm has no regret with respect to the best price on the grid $\mathcal{P}_K$. This is sufficient since the best price in $\mathcal{P}_K$ performs almost as well as the best price in the interval $[0, 1]$, as we formally show in the following lemma.

**Lemma 1.** *It holds that:*
$$\max_{p \in \mathcal{P}_K} g(p) \geq g(p^\star) - {}^L\!/\!_K .$$

*Proof.* Let $k^\star \in [K]$ be such that $p_{k^\star}$ minimizes the distance from $p^\star$ among all the points in $\mathcal{P}_K$. Noticing that a random variable is $1/L$-smooth if an only if its cdf is $L$-Lipschitz continuous, [Cesa-Bianchi et al., 2024b, Lemma 1] ensures that g is $L$-Lipschitz continuous, and hence
$$g(p^\star) - \max_{p \in \mathcal{P}_K} g(p) \leq g(p^\star) - g(p_{k^\star}) \leq L \cdot |p^\star - p_{k^\star}| \leq {}^L\!/\!_K . \qquad \square$$

### 4.2 Define a Clean Event

We now introduce some definitions to aid the presentation. First, we define a set $\mathcal{K}$ (unknown to the learner) that intuitively identifies the prices $p_k \in \mathcal{P}_K$ in which feedback about the seller can be observed with sufficiently high probability, under the asynchronous interaction protocol.

**Definition 1.** *We define $\mathcal{K}$ as the subset of $k \in [K]$ such that:*
$$\mathbb{P}[B \geq p_k] \geq 8T^{-1/3} \log(KT^2/\delta).$$

We will show that for prices in $\mathcal{K}$ our estimates are sufficiently accurate, while prices not in $\mathcal{K}$ can be ignored. Indeed, this is because they achieve a negligible expected gain from trade.

Moreover, we also introduce the following definition of *clean event* $\mathcal{E}$. Intuitively, this is decomposed into four different events, namely $\mathcal{E}_1$, $\mathcal{E}_2$, $\mathcal{E}_3$, and $\mathcal{E}_4$. These events are related to different high-probability concentration bounds. The event $\mathcal{E}_1$ is defined as follows:

$$\mathcal{E}_1 := \bigcap_{t=HK}^{T} \bigcap_{k \in \mathcal{K}} \left\{ \mathcal{Q}_{k,t} > \frac{\mathcal{T}_{k,t}}{2} \mathbb{P}[B \geq p_k] \right\}.$$

Intuitively, under this event, the broker receives feedback about the seller at price $p_k$ at least half of the times the price $p_k$ is proposed, weighted by the probability that the buyer accepts it. It can be shown that this event holds with high probability by the aid of Chernoff's inequality. This event guarantees that we observe a constant fraction of the "expected" samples.

The event $\mathcal{E}_2$ is related to the estimates $\hat{G}_k$ and $\hat{F}_k$ of the integrals appearing in the definition of gain from trade. In particular, it requires that the estimates lie in their respective confidence intervals. Formally, we have:

$$\mathcal{E}_2 := \bigcap_{k \in \mathcal{K}} \left\{ \left| \hat{G}_k - \int_0^{p_k} \mathbb{P}[S \leq \lambda] \, \mathrm{d}\lambda \right| \leq \sqrt{\frac{2 \log(2/\delta)}{K N_k}} + \frac{1}{K} \right\} \cap$$
$$\bigcap_{k \in [K]} \left\{ \left| \hat{F}_k - \int_{p_k}^1 \mathbb{P}[B \geq \lambda] \, \mathrm{d}\lambda \right| \leq \sqrt{\frac{2 \log(2/\delta)}{H K}} + \frac{1}{K} \right\}.$$

The underlying idea is to show that $\hat{G}_k$ is close to its expectation up to $\tilde{\mathcal{O}}\left({1}/{K N_k}\right)$ with high probability due to a combination of Chernoff and Azuma-Hoeffding's inequality. Moreover, the expectation $\mathbb{E}[\hat{G}_k]$ is close to the integral term (d) in eq. (1), by leveraging the rectangle rule to compute integrals and the fact that the integrand function is monotone to obtain telescopic simplifications. An analogous argument holds for the terms $\hat{F}_k$ as well.

Furthermore, $\mathcal{E}_3$ is the event in which the estimates $\hat{\mu}_{k,t}$ and $\hat{\nu}_{k,t}$ of the probabilities of the seller and the buyer accepting the trade, respectively, lie in confidence intervals that shrink as the inverse of the square root of the number of observed samples.

$$\mathcal{E}_3 := \bigcap_{t=HK}^{T} \bigcap_{k \in [K]} \left\{ \hat{\mu}_{k,t} - \sqrt{\frac{\log(2T/\delta)}{2\mathcal{Q}_{k,t}}} \leq \mathbb{P}[S \leq p_k] \leq \hat{\mu}_{k,t} + \sqrt{\frac{\log(2T/\delta)}{2\mathcal{Q}_{k,t}}} \right\} \cap$$
$$\bigcap_{t=HK}^{T} \bigcap_{k \in [K]} \left\{ \hat{\nu}_{k,t} - \sqrt{\frac{\log(2T/\delta)}{2\mathcal{T}_{k,t}}} \leq \mathbb{P}[B \geq p_k] \leq \hat{\nu}_{k,t} + \sqrt{\frac{\log(2T/\delta)}{2\mathcal{T}_{k,t}}} \right\}$$

That $\mathcal{E}_3$ holds with high probability is a consequence of Hoeffding's inequality and a union bound, once we realize that, for each $k \in [K]$, the sequence $(S_{k,j}, B_{k,j})_{j \in \mathbb{N}}$ is i.i.d..

Finally, the event $\mathcal{E}_4$ guarantees that if a price $p_k$ has too low a probability of being accepted by the buyer, i.e., $k \notin \mathcal{K}$, then such a price will not belong to the set $K^\diamond$ of candidate optimal prices used by the algorithm in the second phase. Formally, we have:

$$\mathcal{E}_4 := \bigcap_{k \notin \mathcal{K}} \left\{ \mathcal{Q}_{k,KH} \leq 32 \log\left(\frac{KT^2}{\delta}\right) \right\}.$$

Intuitively, also the probability of event $\mathcal{E}_4$ can be bounded by using Chernoff inequality. This event is important to avoid running the second phase on prices $p_k$ that have too low a probability of being accepted by the buyer. This would result in a small number of samples and huge confidence intervals. Moreover, such prices can be safely discarded as their expected gain from trade is small.

Finally, by bounding each event separately and applying a union bound, we obtain the following lemma, which establishes the probability with which the clean event $\mathcal{E}$ holds.

**Lemma 2.** *Let $\mathcal{E} := \mathcal{E}_1 \cap \mathcal{E}_2 \cap \mathcal{E}_3 \cap \mathcal{E}_4$. Then, we have:*

$$\mathbb{P}[\mathcal{E}] \geq 1 - \mathcal{O}(KT\delta).$$

We defer the formal proof of this lemma to the Appendix.

### 4.3 Bound the Regret

Next, we introduce two crucial lemmas that enable us to derive the regret guarantees of Algorithm 2. Intuitively, the first lemma shows that, under the clean event $\mathcal{E}$, the first term of the upper confidence

bound employed by Algorithm 2 concentrates towards the first term in the gain from trade decomposition, *i.e.*, the product of (a) and (b), at the desired rate. Moreover, it shows that our confidence bounds are always optimistic, an essential requirement for UCB-like algorithms. The second lemma shows an analogous result, but for the second terms.

**Lemma 3.** *Let $k \in \mathcal{K}$. Then, for each $t \geq HK$, conditional on the event $\mathcal{E}$, we have:*

$$0 \leq \left( \hat{\mu}_{k,t} + \sqrt{\frac{\log(2T/\delta)}{2\mathcal{Q}_{k,t}}} \right) \left( \hat{F}_k + \sqrt{\frac{2\log(2/\delta)}{HK}} + \frac{1}{K} \right) - \mathbb{P}[S \leq p_k] \int_{p_k}^{1} \mathbb{P}[B \geq \lambda] \, d\lambda \leq \eta,$$

*where $\eta := C \log\left(\frac{T}{\delta}\right) \left( \frac{1}{T^{1/3}} + \frac{1}{\sqrt{\mathcal{T}_{k,t}}} \right)$ and $C > 0$ is an absolute constant.*

The above lemma shows that the difference between the first component of the confidence bound $UCB_{k,t}$ (defined in eq. (3)) and the first component of the expected gain from trade is proportional to $1/\sqrt{\mathcal{T}_{k,t}}$. At a first glance, this is not what we would expect since the empirical mean $\hat{\mu}_{k,t}$ is estimated using $\mathcal{Q}_{k,t}$ samples. Thus, the confidence bound of $\hat{\mu}_{k,t}$ is proportional to $1/\sqrt{\mathcal{Q}_{k,t}}$. However, such a term is multiplied by (an upper bound of) the probability that the buyer accepts the offer, *i.e.*, $\mathbb{P}[B \geq p_k]$. Thus, if the confidence term is large (*i.e.*, when $\mathcal{Q}_{k,t}$ is small), then the probability that the buyer accepts the trade is low. These two effects compensate for each other, and the resulting contribution ends up scaling as $1/\sqrt{\mathcal{T}_{k,t}}$. We defer the formal proof of this lemma to the Appendix.

**Lemma 4.** *Let $k \in \mathcal{K}$. Then, for each $t \geq HK$, conditional on the event $\mathcal{E}$, we have:*

$$0 \leq \left( \hat{\nu}_{k,t} + \sqrt{\frac{\log(2T/\delta)}{2\mathcal{T}_{k,t}}} \right) \left( \hat{G}_k + \sqrt{\frac{2\log(2/\delta)}{KN_k}} + \frac{1}{K} \right) - \mathbb{P}[B \geq p_k] \int_{0}^{p_k} \mathbb{P}[S \leq \lambda] \, d\lambda \leq \eta,$$

*where $\eta := C \log\left(\frac{T}{\delta}\right) \left( \frac{1}{T^{1/3}} + \frac{1}{\sqrt{\mathcal{T}_{k,t}}} \right)$ and $C > 0$ is an absolute constant.*

Similarly to Lemma 3, the estimate $\hat{G}_k$ is constructed using a number of samples that depends on the probability that the buyer accepts to trade at price $p_k$. This component is *mirrored* with respect to the one in Lemma 3: rather than having limited samples for the probability estimate $\hat{\nu}_{k,t}$, here we may have few samples to compute $\hat{G}_k$, the term that corresponds to the integral estimates. Nonetheless, a similar argument shows that this does not affect the rate at which the confidence bound shrinks, since the confidence interval around $\hat{G}_k$ is scaled by an upper bound on the buyer's acceptance probability—effectively compensating for the lower sample size. We defer the formal proof of this lemma to the Appendix.

We are now ready to present our main result.

**Theorem 1.** *Algorithm 2 guarantees regret $R_T = \tilde{\mathcal{O}}(T^{2/3})$.*

We now present the high-level ideas behind the proof, deferring the formal argument to the Appendix.

To derive regret guarantees for our algorithm, we analyze separately the two phases it executes. The exploration phase always incurs a regret of order $\mathcal{O}(T^{2/3})$, as the exploration strategy is deterministic and runs for $\mathcal{O}(T^{2/3})$ rounds. To provide an upper bound on the regret suffered in the second phase of our algorithm, we consider two cases depending on whether $k^\star$ belongs to $K^\diamond$ or not. If $k^\star \notin K^\diamond$, then we can show that $\mathbb{P}[B \geq p_{k^\star}] = \tilde{\mathcal{O}}(T^{-1/3})$ and, consequently, $\mathrm{g}(p_{k^\star}) = \tilde{\mathcal{O}}(T^{-1/3})$. Therefore, for any possible set $K^\diamond$, the regret suffered by our algorithm is $R_T = \tilde{\mathcal{O}}(T^{2/3})$.

Conversely, if $k^\star \in K^\diamond$, *i.e.*, if $p_{k^\star}$ is not removed before the second phase, then, noticing that $K^\diamond \simeq \mathcal{K}$, Lemmas 3 and 4 guarantee that (i) the confidence bounds of each suboptimal arm scale as $1/\min(T^{1/3}, \sqrt{\mathcal{T}_{k,t}})$, and (ii) all the estimates remain optimistic. By applying a UCB-style analysis and using Lemma 2, we obtain that the cumulative regret in this phase is also of order $\tilde{\mathcal{O}}(T^{2/3})$.

## 5 Conclusions, Limitations, and Future Research

This work introduced a new asynchronous protocol for repeated bilateral trade, where the broker interacts with a seller only after securing agreement from a buyer. Despite the censored nature

of the seller's feedback, we showed that the broker can still achieve the optimal $\tilde{\mathcal{O}}(T^{2/3})$ regret rate previously known only under the richer two-bit feedback model [Cesa-Bianchi et al., 2024a]. Combined with the known impossibility of learning under one-bit feedback [Cesa-Bianchi et al., 2024a], this work suggests that our protocol elicits the minimal amount of information necessary to enable optimal learning.

While our theoretical guarantees are optimal, some limitations suggest interesting directions for future research. First, our analysis focuses on stationary environments. Although adversarial bilateral trade is known to be unlearnable even under full feedback and when competing with the best fixed price in hindsight [Cesa-Bianchi et al., 2024a], it would be valuable to explore intermediate settings lying in between the i.i.d. and fully adversarial dynamics, perhaps by relaxing the notion of learnability to allow for (dynamic) $\alpha$-regret. Another limitation of our work is the absence of contextual information that the broker might observe before posting a price. Extending the framework to contextual settings—where the broker might have access to side information encoding item characteristics, market conditions, or user profiles—remains a challenging and interesting open problem in our censored framework. Finally, following recent works [Bachoc et al., 2024, Cesari and Colomboni, 2025], an intriguing direction for future research is to study alternative objectives, such as *fair gain from trade* or *trading volume*, which reflect different priorities for the broker.

## Acknowledgments and Disclosure of Funding

FB, MC, RC, and AM are partially supported by the FAIR (Future Artificial Intelligence Research) project, funded by the NextGenerationEU program within the PNRR-PE-AI scheme (M4C2, Investment 1.3, Line on Artificial Intelligence) and by the EU Horizon project ELIAS (European Lighthouse of AI for Sustainability, No. 101120237). RC is partially supported by the MIUR PRIN grant 2022EKNE5K (Learning in Markets and Society). AM is partially supported by the Italian MIUR PRIN 2022 project "Targeted Learning Dynamics: Computing Efficient and Fair Equilibria through No-Regret Algorithms".

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

## Omitted Proofs

In order to prove Lemma 2, we need to introduce some auxiliary useful lemmas.

**Lemma 5.** *It holds* $\mathbb{P}[\mathcal{E}_1] \geq 1 - \delta$.

*Proof.* Let $s \geq H$ be an integer, and let $k \in \mathcal{K}$. By the Chernoff bound and the fact that the variables are all i.i.d., we have:

$$\mathbb{P}\left[\sum_{i=1}^{s} \mathbb{I}\{B_{k,i} \geq p_k\} \leq \frac{s}{2}\mathbb{P}[B \geq p_k]\right] \leq e^{-\frac{s\mathbb{P}[B \geq p_k]}{8}}$$

$$= e^{-\frac{s\mathbb{P}[B \geq p_k]\log\left(KT^2/\delta\right)}{8\log\left(KT^2/\delta\right)}}$$

$$= \left(\frac{\delta}{KT^2}\right)^{\frac{s\mathbb{P}[B \geq p_k]}{8\log\left(KT^2/\delta\right)}}$$

$$\leq \left(\frac{\delta}{KT^2}\right)^{\frac{T^{1/3}\mathbb{P}[B \geq p_k]}{8\log\left(KT^2/\delta\right)}} \leq \frac{\delta}{KT^2}.$$

If $t \geq HK$, then $\mathcal{T}_{k,t} \geq H$ according to the definition of $\mathcal{T}_{k,t}$ and how our algorithm works. It follows that

$$\mathbb{P}\left[\bigcup_{t=HK}^{T} \bigcup_{k\in\mathcal{K}} \left\{\mathcal{Q}_{k,t} \leq \frac{\mathcal{T}_{k,t}}{2}\mathbb{P}[B \geq p_k]\right\}\right]$$

$$= \mathbb{P}\left[\bigcup_{s=H}^{T} \bigcup_{t=HK}^{T} \bigcup_{k\in\mathcal{K}} \left\{\mathcal{Q}_{k,t} \leq \frac{\mathcal{T}_{k,t}}{2}\mathbb{P}[B \geq p_k]\right\} \cap \{\mathcal{T}_{k,t} = s\}\right]$$

$$\leq \sum_{s=H}^{T} \sum_{t=HK}^{T} \sum_{k\in\mathcal{K}} \mathbb{P}\left[\left\{\mathcal{Q}_{k,t} \leq \frac{\mathcal{T}_{k,t}}{2}\mathbb{P}[B \geq p_k]\right\} \cap \{\mathcal{T}_{k,t} = s\}\right]$$

$$= \sum_{s=H}^{T} \sum_{t=HK}^{T} \sum_{k\in\mathcal{K}} \mathbb{P}\left[\left\{\sum_{i=1}^{s} \mathbb{I}\{B_{k,i} \geq p_k\} \leq \frac{s}{2}\mathbb{P}[B \geq p_k]\right\} \cap \{\mathcal{T}_{k,t} = s\}\right]$$

$$\leq \sum_{s=H}^{T} \sum_{t=HK}^{T} \sum_{k\in\mathcal{K}} \mathbb{P}\left[\sum_{i=1}^{s} \mathbb{I}\{B_{k,i} \geq p_k\} \leq \frac{s}{2}\mathbb{P}[B \geq p_k]\right]$$

$$\leq \delta.$$

Hence,

$$\mathbb{P}\left[\bigcap_{t=HK}^{T} \bigcap_{k\in\mathcal{K}} \left\{\mathcal{Q}_{k,t} > \frac{\mathcal{T}_{k,t}}{2}\mathbb{P}[B \geq p_k]\right\}\right] \geq 1 - \delta,$$

which concludes the proof. □

**Lemma 6.** *It holds*

$$\mathbb{P}\left[\bigcap_{k\in\mathcal{K}} \left\{\left|\hat{G}_k - \int_0^{p_k} \mathbb{P}[S \leq \lambda]\,\mathrm{d}\lambda\right| \leq \sqrt{\frac{2\log(2/\delta)}{KN_k}} + \frac{1}{K}\right\}\right] \geq 1 - 2K\delta.$$

*Proof.* Under the event $\mathcal{E}_1$, for all $i \in [K]$ such that $\mathbb{P}[B \geq p_i] \geq 8T^{-1/3}\log(KT^2/\delta)$, we have:

$$\mathcal{Q}_{i,HK} \geq \frac{1}{2}\mathcal{T}_{i,HK}\mathbb{P}[B \geq p_i] = \frac{1}{2}H\mathbb{P}[B \geq p_i],$$

since $\mathcal{T}_{i,HK} = H$ for every $i \in [K]$, according how our algorithm works. This implies that, under the event $\mathcal{E}_1$, the following holds:

$$N_k = \min_{i\leq k} \mathcal{Q}_{i,HK} \geq \frac{1}{2}\min_{i\leq k} H\mathbb{P}[B \geq p_i] = \frac{1}{2}H\mathbb{P}[B \geq p_k] := n_k.$$

As a first step, we prove that, conditioned to the event $\{N_k = \ell\} \cap \mathcal{E}_1$, the estimates $\hat{G}_k$ concentrate around their expectation. Specifically, whenever $\ell \geq n_k$, the following holds:

$$\mathbb{P}\left[\left|\hat{G}_k - \mathbb{E}[\hat{G}_k]\right| \leq 2\sqrt{\frac{\log(2/\delta)}{K\ell}} \,\Big|\, \{N_k = \ell\} \cap \mathcal{E}_1\right] \geq 1 - \delta. \tag{4}$$

Indeed, by Azuma–Hoeffding inequality, we have:

$$\mathbb{P}\left[\left|\sum_{i=1}^{k}\sum_{j=1}^{\ell}\mathbb{I}\{S_{i,j} \leq p_i\} - \sum_{i=1}^{k}\sum_{j=1}^{\ell}\mathbb{P}[S \leq p_i]\right| \leq \sqrt{2K\ell\log(2/\delta)}\right] \geq 1 - \delta .$$

Furthermore, noticing that

$$\left|\hat{G}_k - \mathbb{E}[\hat{G}_k]\right| = \frac{1}{K\ell}\left|\sum_{i=1}^{k}\sum_{j=1}^{\ell}\mathbb{I}\{S_{i,j} \leq p_i\} - \sum_{i=1}^{k}\sum_{j=1}^{\ell}\mathbb{P}[S \leq p_i]\right| ,$$

and that $\{N_k = \ell\} \cap \mathcal{E}_1$ is $\mathbb{P}$-independent from $S_1, S_2, \ldots$, we have:

$$\mathbb{P}\left[\left|\hat{G}_k - \mathbb{E}[\hat{G}_k]\right| \leq \sqrt{\frac{2\log(2/\delta)}{K\ell}} \,\Big|\, \{N_k = \ell\} \cap \mathcal{E}_1\right] = \mathbb{P}\left[\left|\hat{G}_k - \mathbb{E}[\hat{G}_k]\right| \leq \sqrt{\frac{2\log(2/\delta)}{K\ell}}\right]$$
$$\geq 1 - \delta ,$$

showing that Equation 4 holds. Therefore, we can prove that:

$$\mathbb{P}\left[\left|\hat{G}_k - \mathbb{E}[\hat{G}_k]\right| \leq \sqrt{\frac{2\log(2/\delta)}{KN_k}} \,\Big|\, \mathcal{E}_1\right]$$
$$\geq \sum_{\substack{\ell=0,\\\ell\geq n_k}}^{H}\mathbb{P}\left[\left|\hat{G}_k - \mathbb{E}[\hat{G}_k]\right| \leq \sqrt{\frac{2\log(2/\delta)}{K\ell}} \,\Big|\, \{N_k = \ell\} \cap \mathcal{E}_1\right]\mathbb{P}\left[N_k = \ell \mid \mathcal{E}_1\right]$$
$$\geq \sum_{\substack{\ell=0,\\\ell\geq n_k}}^{H}(1-\delta)\mathbb{P}\left[N_k = \ell \mid \mathcal{E}_1\right]$$
$$= 1 - \delta,$$

where the first inequality holds because of the law of total probability, noticing that $\mathbb{P}[N_k = \ell \mid \mathcal{E}_1] = 0$ for all $\ell < n_k$, while the second inequality by eq. (4). Thanks to the i.i.d. hypothesis we have:

$$\mathbb{E}[\hat{G}_k] - \int_0^{p_k}\mathbb{P}[S \leq \lambda]\,\mathrm{d}\lambda = \mathbb{E}\left[\frac{1}{KN_k}\sum_{i=1}^{k}\sum_{j=1}^{N_k}\mathbb{I}\{S_{i,j} \leq p_i\}\right] - \int_0^{p_k}\mathbb{P}[S \leq \lambda]\,\mathrm{d}\lambda$$
$$= \frac{1}{K}\sum_{i=1}^{k}\mathbb{P}[S \leq p_i] - \int_0^{p_k}\mathbb{P}[S \leq \lambda]\,\mathrm{d}\lambda$$
$$= \sum_{i=1}^{k}\int_{\frac{i-1}{K}}^{\frac{i}{K}}\left(\mathbb{P}\left[S \leq \frac{i}{K}\right] - \mathbb{P}[S \leq \lambda]\right)\mathrm{d}\lambda =: (\star) ,$$

Now, due to the fact that $\lambda \mapsto \mathbb{P}[S \leq \lambda]$ is a non-decreasing function, we have that

$$0 \leq (\star) \leq \sum_{i=1}^{k}\int_{\frac{i-1}{K}}^{\frac{i}{K}}\left(\mathbb{P}\left[S \leq \frac{i}{K}\right] - \mathbb{P}\left[S \leq \frac{i-1}{K}\right]\right)\mathrm{d}\lambda$$
$$= \frac{1}{K}\sum_{i=1}^{k}\left(\mathbb{P}\left[S \leq \frac{i}{K}\right] - \mathbb{P}\left[S \leq \frac{i-1}{K}\right]\right) = \frac{\mathbb{P}[S \leq p_k] - \mathbb{P}[S \leq 0]}{K} \leq \frac{1}{K}.$$

Thus, the following holds:

$$\mathbb{P}\left[\left|\hat{G}_k - \int_0^{p_k} \mathbb{P}[S \le \lambda]\, d\lambda\right| \le \sqrt{\frac{2\log(2/\delta)}{KN_k}} + \frac{1}{K} \,\middle|\, \mathcal{E}_1\right] \ge 1 - \delta.$$

Thus, thanks to Lemma 5, we have:

$$\mathbb{P}\left[\left|\hat{G}_k - \int_0^{p_k} \mathbb{P}[S \le \lambda]\, d\lambda\right| \le \sqrt{\frac{2\log(2/\delta)}{KN_k}} + \frac{1}{K}\right]$$

$$\ge \mathbb{P}\left[\left|\hat{G}_k - \int_0^{p_k} \mathbb{P}[S \le \lambda]\, d\lambda\right| \le \sqrt{\frac{2\log(2/\delta)}{KN_k}} + \frac{1}{K} \,\middle|\, \mathcal{E}_1\right]\mathbb{P}[\mathcal{E}_1]$$

$$\ge 1 - 2\delta.$$

Finally, taking a union bound over all possible sets of arms $\mathcal{K}$, we prove the lemma. $\qquad\square$

**Lemma 7.** *It holds*

$$\mathbb{P}\left[\bigcap_{k \in \mathcal{K}}\left\{\left|\hat{F}_k - \int_{p_k}^1 \mathbb{P}[B \ge \lambda]\, d\lambda\right| \le \sqrt{\frac{2\log(2/\delta)}{KH}} + \frac{1}{K}\right\}\right] \ge 1 - K\delta.$$

*Proof.* Thanks to the i.i.d. hypothesis and the fact that $\lambda \mapsto \mathbb{P}[B \ge \lambda]$ is a non-increasing function, with an argument analogous to that provided in the proof of Lemma 6, we have that

$$\left|\mathbb{E}[\hat{F}_k] - \int_{p_k}^1 \mathbb{P}[B \ge \lambda]\, d\lambda\right| \le \frac{1}{K}.$$

By Azuma–Hoeffding inequality, we have:

$$\left|\sum_{i=1}^k \sum_{j=1}^H \mathbb{I}\{B_{i,j} \ge p_i\} - \sum_{i=1}^k \sum_{j=1}^H \mathbb{P}[B \ge p_i]\right| \le \sqrt{2kH\log(2/\delta)} \le \sqrt{2KH\log(2/\delta)}$$

with probability $1 - \delta$. Hence, to conclude, it is enough to notice that

$$\left|\hat{F}_k - \mathbb{E}[\hat{F}_k]\right| = \frac{1}{KH}\left|\sum_{i=1}^k \sum_{j=1}^H \mathbb{I}\{B_{i,j} \ge p_i\} - \sum_{i=1}^k \sum_{j=1}^H \mathbb{P}[B \ge p_i]\right|$$

and take a union bound over all possible sets of arms $[K]$. $\qquad\square$

**Lemma 8.** *It holds* $\mathbb{P}[\mathcal{E}_3] \ge 1 - 2KT\delta$.

*Proof.* Fix $k \in \mathcal{K}$ and $t \ge HK$. Recall that, by definition,

$$\hat{\mu}_{k,t} := \frac{\sum_{\tau=1}^{\mathcal{Q}_{k,t}} \mathbb{I}\{S_{k,\tau} \le p_k\}}{\mathcal{Q}_{k,t}}.$$

Employing a union bound and the Hoeffding inequality, we have that:

$$\hat{\mu}_{k,t} - \sqrt{\frac{\log(2T/\delta)}{2\mathcal{Q}_{k,t}}} \le \mathbb{P}[S \le p_k] \le \hat{\mu}_{k,t} + \sqrt{\frac{\log(2T/\delta)}{2\mathcal{Q}_{k,t}}}$$

with probability at least $1 - \delta$. Thus, taking a union bound, we have:

$$\mathbb{P}\left[\bigcap_{k \in [K]}\bigcap_{t=HK}^T\left\{\hat{\mu}_{k,t} - \sqrt{\frac{\log(2T/\delta)}{2\mathcal{Q}_{k,t}}} \le \mathbb{P}[S \le p_k] \le \hat{\mu}_{k,t} + \sqrt{\frac{\log(2T/\delta)}{2\mathcal{Q}_{k,t}}}\right\}\right] \ge 1 - KT\delta.$$

With an analogous argument, it is possible to show that:

$$\mathbb{P}\left[\bigcap_{k \in [K]}\bigcap_{t=HK}^T\left\{\hat{\nu}_{k,t} - \sqrt{\frac{\log(2T/\delta)}{2\mathcal{T}_{k,t}}} \le \mathbb{P}[B \ge p_k] \le \hat{\nu}_{k,t} + \sqrt{\frac{\log(2T/\delta)}{2\mathcal{T}_{k,t}}}\right\}\right] \ge 1 - KT\delta.$$

Thus, taking a union bound, we have that the lemma holds. $\qquad\square$

**Lemma 9.** *It holds* $\mathbb{P}[\mathcal{E}_4] \geq 1 - \delta$

*Proof.* Let $\epsilon = 8T^{-1/3}\log(KT^2/\delta)$ and $\hat{\nu}_k := \hat{\nu}_{k,KH}$. If $k \notin \mathcal{K}$, then $\mathbb{P}[B \geq p_k] \leq \epsilon$. Therefore, we can employ the multiplicative Chernoff inequality as follows:

$$\mathbb{P}\big[\hat{\nu}_k \geq (1+c)\mathbb{P}[B \geq p_k]\big] \leq e^{-\frac{c^2 H\mathbb{P}[B \geq p_k]}{2+c}},$$

with $c = \frac{\epsilon}{\mathbb{P}[B \geq p_k]}$. Thus, we get:

$$\mathbb{P}\big[\hat{\nu}_k \geq \mathbb{P}[B \geq p_k] + \epsilon\big] \leq \exp\left(-\frac{\left(\frac{\epsilon}{\mathbb{P}[B \geq p_k]}\right)^2 H\mathbb{P}[B \geq p_k]}{2 + \frac{\epsilon}{\mathbb{P}[B \geq p_k]}}\right)$$

$$\leq \exp\left(-\frac{\frac{\epsilon}{\mathbb{P}[B \geq p_k]}}{2 + \frac{\epsilon}{\mathbb{P}[B \geq p_k]}} \cdot \epsilon H\right)$$

$$\leq \left(\frac{\delta}{KT^2}\right)^{8/3} \leq \frac{\delta}{KT^2},$$

since $x/(x+2) \geq 1/3$, for every $x \geq 1$. As a result, we have:

$$\mathbb{P}\big[\mathcal{Q}_{k,HK} \leq 2H\epsilon\big] = \mathbb{P}\big[\hat{\nu}_k \leq 2\epsilon\big] \geq \mathbb{P}\big[\hat{\nu}_k \leq \mathbb{P}[B \geq p_k] + \epsilon\big] \geq 1 - \frac{\delta}{KT^2},$$

recalling that $\hat{\nu}_{k,HK} = \mathcal{Q}_{k,HK}/H$. Furthermore, we notice that:

$$2H\epsilon \leq 16HT^{-1/3}\log(KT^2/\delta) = 16\frac{\lceil T^{1/3}\rceil}{T^{1/3}}\log(KT^2/\delta) \leq 32\log(KT^2/\delta).$$

Thus, by taking a union bound, we have:

$$\mathbb{P}\left[\bigcap_{k \notin \mathcal{K}} \left\{\mathcal{Q}_{k,HK} \leq 32\log(KT^2/\delta)\right\}\right] \geq 1 - \frac{\delta}{T^2} \geq 1 - \delta,$$

concluding the proof. $\qquad\square$

**Lemma 2.** *Let $\mathcal{E} := \mathcal{E}_1 \cap \mathcal{E}_2 \cap \mathcal{E}_3 \cap \mathcal{E}_4$. Then, we have:*
$$\mathbb{P}[\mathcal{E}] \geq 1 - \mathcal{O}(KT\delta).$$

*Proof.* Thanks to Lemmas 5, 6, 7, 8, and 9, by taking a union bound we have:
$$\mathbb{P}[\mathcal{E}] \geq 1 - (5KT + 2)\delta = 1 - \mathcal{O}(KT\delta),$$

concluding the proof. $\qquad\square$

**Lemma 3.** *Let $k \in \mathcal{K}$. Then, for each $t \geq HK$, conditional on the event $\mathcal{E}$, we have:*

$$0 \leq \left(\hat{\mu}_{k,t} + \sqrt{\frac{\log(2T/\delta)}{2\mathcal{Q}_{k,t}}}\right)\left(\hat{F}_k + \sqrt{\frac{2\log(2/\delta)}{HK}} + \frac{1}{K}\right) - \mathbb{P}[S \leq p_k]\int_{p_k}^{1}\mathbb{P}[B \geq \lambda]\,d\lambda \leq \eta,$$

*where $\eta := C\log\left(\frac{T}{\delta}\right)\left(\frac{1}{T^{1/3}} + \frac{1}{\sqrt{\mathcal{T}_{k,t}}}\right)$ and $C > 0$ is an absolute constant.*

*Proof.* We first prove that:

$$\left(\hat{\mu}_{k,t} + \sqrt{\frac{\log(2T/\delta)}{2\mathcal{Q}_{k,t}}}\right)\left(\hat{F}_k + \sqrt{\frac{2\log(2/\delta)}{HK}} + \frac{1}{K}\right)$$

is the (optimistic) estimator of

$$\mathbb{P}[S \leq p_k]\int_{p_k}^{1}\mathbb{P}[B \geq \lambda]\,d\lambda.$$

To do so, we observe that under the event $\mathcal{E}$, we have:

$$\left(\hat{\mu}_{k,t} + \sqrt{\frac{\log(2T/\delta)}{2\mathcal{Q}_{k,t}}}\right)\left(\hat{F}_k + \sqrt{\frac{2\log(2/\delta)}{HK}} + \frac{1}{K}\right) - \mathbb{P}[S \le p_k]\int_{p_k}^1 \mathbb{P}[B \ge \lambda]\,\mathrm{d}\lambda$$

$$\le \left(\mathbb{P}[S \le p_k] + \sqrt{\frac{2\log(2T/\delta)}{\mathcal{Q}_{k,t}}}\right)\left(\int_{p_k}^1 \mathbb{P}[B \ge \lambda]\,\mathrm{d}\lambda + \sqrt{\frac{8\log(2/\delta)}{HK}} + \frac{2}{K}\right)$$

$$- \mathbb{P}[S \le p_k]\int_{p_k}^1 \mathbb{P}[B \ge \lambda]\,\mathrm{d}\lambda$$

$$\le \sqrt{\frac{2\log(2T/\delta)}{\mathcal{Q}_{k,t}}}\int_{p_k}^1 \mathbb{P}[B \ge \lambda]\,\mathrm{d}\lambda + \mathbb{P}[S \le p_k]\sqrt{\frac{8\log(2/\delta)}{HK}} + \frac{4\log(2T/\delta)}{\sqrt{HK\mathcal{Q}_{k,t}}}$$

$$+ \frac{2}{K}\left(\mathbb{P}[S \le p_k] + \sqrt{\frac{2\log(2T/\delta)}{\mathcal{Q}_{k,t}}}\right)$$

$$\le \underbrace{\sqrt{\frac{2\log(2T/\delta)}{\mathcal{Q}_{k,t}}}\mathbb{P}[B \ge p_k]}_{(\star)} + \frac{20\log(2T/\delta)}{T^{1/3}}.$$

Now, notice that, since $k \in \mathcal{K}$ by assumption, we have $\mathcal{Q}_{k,t} \ge \frac{1}{2}\mathcal{T}_{k,t}\mathbb{P}[B \ge p_k]$ for all $t \ge HK$, under the event $\mathcal{E}$. Therefore, the following holds:

$$(\star) = \sqrt{\frac{2\log(2T/\delta)}{\mathcal{Q}_{k,t}}}\mathbb{P}[B \ge p_k] \le \sqrt{\frac{4\log(2T/\delta)}{\mathcal{T}_{k,t}\mathbb{P}[B \ge p_k]}}\mathbb{P}[B \ge p_k] \le 2\frac{\log(2T/\delta)}{\sqrt{\mathcal{T}_{k,t}}}.$$

Putting all together, we have:

$$\left(\hat{\mu}_{k,t} + \sqrt{\frac{\log(2T/\delta)}{2\mathcal{Q}_{k,t}}}\right)\left(\hat{F}_k + \sqrt{\frac{2\log(2/\delta)}{HK}} + \frac{1}{K}\right) - \mathbb{P}[S \le p_k]\int_{p_k}^1 \mathbb{P}[B \ge \lambda]\,\mathrm{d}\lambda$$

$$\le \log\left(\frac{T}{\delta}\right)\mathcal{O}\left(\frac{1}{\sqrt{\mathcal{T}_{k,t}}} + \frac{1}{T^{1/3}}\right).$$

Finally, we notice that:

$$\mathbb{P}[S \le p_k]\int_{p_k}^1 \mathbb{P}[B \ge \lambda]\,\mathrm{d}\lambda \le \left(\hat{\mu}_{k,t} + \sqrt{\frac{\log(2T/\delta)}{2\mathcal{Q}_{k,t}}}\right)\left(\hat{F}_k + \sqrt{\frac{2\log(2/\delta)}{HK}} + \frac{1}{K}\right),$$

as a direct consequence of being under the clean event $\mathcal{E}$. This concludes the proof. $\square$

**Lemma 4.** *Let $k \in \mathcal{K}$. Then, for each $t \ge HK$, conditional on the event $\mathcal{E}$, we have:*

$$0 \le \left(\hat{\nu}_{k,t} + \sqrt{\frac{\log(2T/\delta)}{2\mathcal{T}_{k,t}}}\right)\left(\hat{G}_k + \sqrt{\frac{2\log(2/\delta)}{KN_k}} + \frac{1}{K}\right) - \mathbb{P}[B \ge p_k]\int_0^{p_k} \mathbb{P}[S \le \lambda]\,\mathrm{d}\lambda \le \eta,$$

*where $\eta := C\log\left(\frac{T}{\delta}\right)\left(\frac{1}{T^{1/3}} + \frac{1}{\sqrt{\mathcal{T}_{k,t}}}\right)$ and $C > 0$ is an absolute constant.*

*Proof.* Notice that the first inequality is trivially given by the fact that, under the event $\mathcal{E}$, the quantity

$$\left(\hat{\nu}_{k,t} + \sqrt{\frac{\log(2T/\delta)}{2\mathcal{T}_{k,t}}}\right)\left(\hat{G}_k + \sqrt{\frac{2\log(2/\delta)}{KN_k}} + \frac{1}{K}\right)$$

is an (optimistic) estimator of

$$\mathbb{P}[B \ge p_k]\int_0^{p_k} \mathbb{P}[S \le \lambda]\,\mathrm{d}\lambda.$$

For the second inequality, we notice that, under the event $\mathcal{E}$:

$$\left(\hat{\nu}_{k,t} + \sqrt{\frac{\log(2T/\delta)}{2\mathcal{T}_{k,t}}}\right)\left(\hat{G}_k + \sqrt{\frac{2\log(2/\delta)}{KN_k}} + \frac{1}{K}\right) - \mathbb{P}[B \geq p_k]\int_0^{p_k}\mathbb{P}[S \leq \lambda]\,\mathrm{d}\lambda$$

$$\overset{\mathcal{E}\subset\mathcal{E}_2\cap\mathcal{E}_3}{\leq} \left(\mathbb{P}[B \geq p_k] + \sqrt{\frac{2\log(2T/\delta)}{\mathcal{T}_{k,t}}}\right)\left(\int_0^{p_k}\mathbb{P}[S \leq \lambda]\,\mathrm{d}\lambda + \sqrt{\frac{8\log(2/\delta)}{KN_k}} + \frac{2}{K}\right)$$

$$- \mathbb{P}[B \geq p_k]\int_0^{p_k}\mathbb{P}[S \leq \lambda]\,\mathrm{d}\lambda$$

$$\overset{\mathcal{E}\subset\mathcal{E}_1}{\leq} \sqrt{\frac{2\log(2T/\delta)}{\mathcal{T}_{k,t}}}\int_0^{p_k}\mathbb{P}[S \leq \lambda]\,\mathrm{d}\lambda + 4\mathbb{P}[B \geq p_k]\sqrt{\frac{\log(2/\delta)}{KH\mathbb{P}[B \geq p_k]}}$$

$$+ \frac{4\sqrt{2}\log(2T/\delta)}{\sqrt{KH\mathbb{P}[B \geq p_k]}\sqrt{\mathcal{T}_{k,t}}} + \frac{2}{K}\left(\mathbb{P}[B \geq p_k] + \sqrt{\frac{2\log(2T/\delta)}{\mathcal{T}_{k,t}}}\right)$$

$$\overset{k\in\mathcal{K}}{=} \log\left(\frac{T}{\delta}\right)\cdot\mathcal{O}\left(\frac{1}{\sqrt{\mathcal{T}_{k,t}}} + \frac{1}{T^{1/3}}\right),$$

concluding the proof. $\square$

**Theorem 1.** *Algorithm 2 guarantees regret* $R_T = \tilde{\mathcal{O}}(T^{2/3})$.

*Proof.* We first notice that, by defining

$$\mathcal{R}_T := \sum_{t=1}^T \mathrm{g}(p^\star) - \sum_{t=1}^T \mathrm{g}(P_t),$$

we have that $R_T = \mathbb{E}[\mathcal{R}_T]$ and

$$R_T = \mathbb{E}[\mathcal{R}_T\mathbb{I}_{\mathcal{E}}] + \mathbb{E}[\mathcal{R}_T\mathbb{I}_{\mathcal{E}^c}]$$
$$\leq \mathbb{E}[\mathcal{R}_T\mathbb{I}_{\mathcal{E}}] + \mathbb{E}[T\mathbb{I}_{\mathcal{E}^c}]$$
$$\leq \mathbb{E}[\mathcal{R}_T\mathbb{I}_{\mathcal{E}}] + 6KT^2\delta = \mathbb{E}[\mathcal{R}_T\mathbb{I}_{\mathcal{E}}] + \mathcal{O}(T^{1/3}).$$

It is then sufficient to control the magnitude of $\mathcal{R}_T$ under the clean event $\mathcal{E}$. Hence, from this point on, we assume we are under the clean event $\mathcal{E}$.

Let $k^\star \in \arg\max_{k\in[K]}\mathrm{g}(p_k)$.

First, notice that, if $\mathbb{P}[p_{k^\star} \leq B] \leq 64T^{-1/3}\log(KT^2/\delta)$, then

$$\mathrm{g}(p_{k^\star}) = \mathbb{E}[(B - S)\,\mathbb{I}\{S \leq p_{k^\star} \leq B\}] \leq \mathbb{E}[\mathbb{I}\{B \geq p_{k^\star}\}] = \mathbb{P}[B \geq p_{k^\star}] \leq 64T^{-1/3}\log(KT^2/\delta),$$

where we used $(B - S) \leq 1$ and $\{S \leq p \leq B\} \subseteq \{B \geq p\}$. Thus, when $\mathbb{P}[p_{k^\star} \leq B] \leq 64T^{-1/3}\log(KT^2/\delta)$, we have, due to Lemma 1, that if we pay an additional term whose instantaneous regret is upper bounded by $L/K$, we can control $\mathcal{R}_T$ by comparing our performance against the performance of the best point in the grid $p_{k^\star}$, from which

$$\mathcal{R}_T = T\cdot\tilde{\mathcal{O}}(T^{-1/3}) + \frac{TL}{K} = \tilde{\mathcal{O}}(T^{2/3}).$$

Hence, we are left to analyze what happens when $\mathbb{P}[p_{k^\star} \leq B] > 64T^{-1/3}\log(KT^2/\delta)$, which we assume being the case from this point on. First, since $\mathbb{P}[p_{k^\star} \leq B] > 64T^{-1/3}\log(KT^2/\delta)$, given that $\mathcal{E} \subset \mathcal{E}_1$, it follows that $k^\star \in K^\diamond$.

We now notice that for each $k \in K^\diamond$ we have that $\mathcal{Q}_{k,HK} > 32T^{-1/3}\log(KT^2/\delta)$ by definition. In the clean event $\mathcal{E}$, we have that $\mathcal{E}_4$ holds, and hence for each $h \notin \mathcal{K}$ we have that $\mathcal{Q}_{h,HK} \leq 32T^{-1/3}\log(KT^2/\delta)$. It follows that, in the clean event $\mathcal{E}$, $k \in K^\diamond$ implies $k \in \mathcal{K}$, *i.e.*, $K^\diamond \subset \mathcal{K}$.

Now, we recall that Lemma 3 and Lemma 4 imply that, under the event $\mathcal{E}$, for all $t \geq HK + 1$ and $k \in \mathcal{K}$:

$$g(p_k) \leq UCB_{k,t-1} \leq g(p_k) + \eta_{k,t-1} \, , \tag{5}$$

where $\eta_{k,t-1} := \tilde{C} \log\left(\frac{T}{\delta}\right)\left(\frac{1}{T^{1/3}} + \frac{1}{\sqrt{\mathcal{T}_{k,t-1}}}\right)$ and $\tilde{C} > 0$ is a universal constant. (We use $UCB_{k,t-1}$ because $P_t$ is chosen at the start of round $t$ based only on information up to time $t-1$.)

If, for every $p \in [0,1]$, we define the quantity $\Delta_p := g(p_{k^\star}) - g(p)$, then, for each $t \geq HK + 1$, if $k_t \in K^\diamond$ is such that $P_t = p_{k_t}$, by eq. (5) we have

$$g(p_{k^\star}) = \max_{k \in K^\diamond} g(p_k) \leq \max_{k \in K^\diamond} UCB_{k,t-1} = UCB_{k_t,t-1} \leq g(P_t) + \eta_{k_t,t-1} \, ,$$

and hence

$$\Delta_{P_t} \leq \eta_{k_t,t-1} \, .$$

In addition, by Lemma 1 and the fact that the instantaneous regret is upper bounded by 1, we have:

$$\mathcal{R}_T \leq HK + \frac{LT}{K} + \sum_{t=HK+1}^{T} \Delta_{P_t}.$$

Now, we have

$$\sum_{t=HK+1}^{T} \Delta_{P_t} = \sum_{k \in K^\diamond} \sum_{t=HK+1}^{T} \Delta_{P_t} \mathbb{I}\{P_t = p_k\}$$

$$\leq \sum_{k \in K^\diamond} \sum_{t=HK+1}^{T} \eta_{k,t-1} \mathbb{I}\{P_t = p_k\}$$

$$\leq \sum_{k \in K^\diamond} \sum_{t=HK+1}^{T} \left[\tilde{C} \log\left(\frac{T}{\delta}\right)\left(\frac{1}{T^{1/3}} + \frac{1}{\sqrt{\mathcal{T}_{k,t-1}}}\right)\right] \mathbb{I}\{P_t = p_k\} \quad \text{by the definition of } \eta_{k,t-1}$$

$$= \tilde{C} \log\left(\frac{T}{\delta}\right)\left[\frac{1}{T^{1/3}} \sum_{k \in K^\diamond} \sum_{t=HK+1}^{T} \mathbb{I}\{P_t = p_k\} + \sum_{k \in K^\diamond} \sum_{t=HK+1}^{T} \frac{\mathbb{I}\{P_t = p_k\}}{\sqrt{\mathcal{T}_{k,t-1}}}\right]$$

$$\leq \tilde{C} \log\left(\frac{T}{\delta}\right)\left[\frac{1}{T^{1/3}} \sum_{t=HK+1}^{T} 1 + \sum_{k \in K^\diamond} \sum_{t=HK+1}^{T} \frac{\mathbb{I}\{P_t = p_k\}}{\sqrt{\mathcal{T}_{k,t-1}}}\right]$$

$$\leq \tilde{C} \log\left(\frac{T}{\delta}\right)\left[T^{2/3} + \sum_{k \in K^\diamond} \sum_{t=HK+1}^{T} \frac{\mathbb{I}\{P_t = p_k\}}{\sqrt{\mathcal{T}_{k,t-1}}}\right]$$

$$\leq \tilde{C} \log\left(\frac{T}{\delta}\right)\left[T^{2/3} + \sum_{k \in K^\diamond} \underbrace{\sum_{t=HK+1}^{T} \frac{\mathbb{I}\{P_t = p_k\}}{\sqrt{\mathcal{T}_{k,t-1}}}}_{\leq \, 2\sqrt{\mathcal{T}_{k,T}} \text{ by (a)}}\right]$$

$$\leq \tilde{C} \log\left(\frac{T}{\delta}\right)\left[T^{2/3} + 2 \sum_{k \in K^\diamond} \sqrt{\mathcal{T}_{k,T}}\right]$$

$$\leq \tilde{C} \log\left(\frac{T}{\delta}\right)\left[T^{2/3} + 2\sqrt{|K^\diamond| \sum_{k \in K^\diamond} \mathcal{T}_{k,T}}\right] \quad \text{(b) Cauchy–Schwarz}$$

$$\leq \tilde{C} \log\left(\frac{T}{\delta}\right)\left[T^{2/3} + 2\sqrt{KT}\right]$$

$$= \tilde{\mathcal{O}}(T^{2/3}),$$

where (a) and (b) can be proved as follows

(a) $\displaystyle\sum_{t=HK+1}^{T} \frac{\mathbb{I}\{P_t = p_k\}}{\sqrt{\mathcal{T}_{k,t-1}}} = \sum_{i=1}^{m_k} \frac{1}{\sqrt{H+(i-1)}}$ where $m_k \coloneqq \displaystyle\sum_{t=HK+1}^{T} \mathbb{I}\{P_t = p_k\},\ \ H = \lceil T^{1/3}\rceil$

$$\leq \sum_{j=H}^{H+m_k-1} \frac{1}{\sqrt{j}} \leq \int_{H-1}^{H+m_k-1} \frac{1}{\sqrt{x}}\,\mathrm{d}x = 2\big(\sqrt{H+m_k-1} - \sqrt{H-1}\big)$$

$$\leq 2\sqrt{m_k} \leq 2\sqrt{\mathcal{T}_{k,T}},$$

(b) $\displaystyle\sum_{k\in K^\diamond} \sqrt{\mathcal{T}_{k,T}} \leq \sqrt{|K^\diamond| \sum_{k\in K^\diamond} \mathcal{T}_{k,T}} \leq \sqrt{KT},$ since $\displaystyle\sum_{k\in K^\diamond} \mathcal{T}_{k,T} \leq \sum_{t=1}^{T}\sum_k \mathbb{I}\{P_t = p_k\} = T.$

Hence

$$\mathbb{E}[\mathcal{R}_T \mathbb{I}_\mathcal{E}] \leq \tilde{C}' \log\left(\frac{T}{\delta}\right) T^{2/3} = \tilde{\mathcal{O}}(T^{2/3}),$$

concluding the proof. □

## Experimental Results

In this section, we present some experimental results obtained on synthetically-generated instances. Specifically, we consider instances where the seller's valuations are sampled from a Beta distribution with parameters $\alpha_s$ and $\beta_s$, while the buyer's valuations are sampled from a Beta distribution with parameters $\alpha_b$ and $\beta_b$. For each instance, we evaluate the performance of our algorithm and the Scouting Bandits algorithm of Cesa-Bianchi et al. [2021] in terms of cumulative regret. To this end, we run both algorithms on each instance $n = 5$ times and report the mean and standard deviation of the achieved cumulative regret.

Table 1: Comparison between our algorithm and the one of Cesa-Bianchi et al. [2021] in terms of cumulative regret across different instances where buyers and sellers' valuations are distributed according to Beta distributions.

| Parameter | Instance 1 | Instance 2 | Instance 3 |
|---|---|---|---|
| Time horizon ($T$) | 10000 | 50000 | 10000 |
| ($\alpha_s, \beta_s$) | (5.0, 10.0) | (5.0, 10.0) | (10.0, 10.0) |
| ($\alpha_b, \beta_b$) | (15.0, 10.0) | (15.0, 10.0) | (15.0, 10.0) |
| **Regret $\pm$ std (ours)** | $199.6 \pm 17.1$ | $714.4 \pm 73.1$ | $135.2 \pm 22.1$ |
| Regret $\pm$ std (Cesa-Bianchi et al. [2021]) | $732.0 \pm 21.4$ | $2253.8 \pm 75.2$ | $548.8 \pm 16.4$ |

Table 2: Comparison between our algorithm and the one of Cesa-Bianchi et al. [2021] in terms of cumulative regret across different instances where buyers and sellers' valuations are distributed according to Beta distributions.

| Parameter | Instance 4 | Instance 5 | Instance 6 |
|---|---|---|---|
| Time horizon ($T$) | 50000 | 10000 | 50000 |
| ($\alpha_s, \beta_s$) | (10.0, 10.0) | (2.0, 3.0) | (2.0, 3.0) |
| ($\alpha_b, \beta_b$) | (15.0, 10.0) | (10.0, 10.0) | (10.0, 10.0) |
| **Regret $\pm$ std (ours)** | $326.5 \pm 84.0$ | $168.5 \pm 28.8$ | $439.2 \pm 136.2$ |
| Regret $\pm$ std (Cesa-Bianchi et al. [2021]) | $2381.0 \pm 98.1$ | $583.5 \pm 27.6$ | $2502.4 \pm 52.2$ |

We observe that the regret incurred by our algorithm is lower than that of Cesa-Bianchi et al. [2021]. While this may appear counterintuitive (since we use less feedback than the Scouting Bandits algorithm of Cesa-Bianchi et al. [2021]), the improvement stems from a key difference. Indeed, after the initial exploration phase, we eliminate arms that are guaranteed to be suboptimal. This pruning

step, absent in Scouting Bandits, allows us to restrict the subsequent bandit phase to the reduced set $K^\diamond$, which can be significantly smaller than the original set of arms. In contrast, Cesa-Bianchi et al. [2021] run their algorithm over the full set, whose size is $T^{1/3}$.

