# OpenReview forum: "Online Bilateral Trade With Minimal Feedback: Don’t Waste Seller’s Time"
_NeurIPS.cc/2025/Conference — NeurIPS 2025 poster_

### Official Review · Reviewer_vJmF · 2025-06-04

**Clarity:** 4
**Significance:** 4
**Originality:** 3
**Rating:** 5
**Confidence:** 3

**Summary:**

Bilateral trading refers to the situation in which a buyer and a seller wish to trade an asset through an intermediary. The goal of the intermediary is to post a price that maximizes gains from trade (or, equivalently, total welfare). In an online learning setting, the price is chosen dynamically based on the feedback received by the intermediary at the end of each trading round. In the one-bit model, the intermediary only observes whether the trade has occurred or not. However, it is known that this is insufficient for learning optimal price-setting strategies. In the two-bit model, the intermediary simultaneously observes whether the buyer and the seller are willing to trade. In this case, it is known that the bound on the regret is O(T^{2/3}). The contribution of this paper is to show that the same bound can be achieved under weaker feedback (3/2-bit, roughly speaking). The paper presents an algorithm where the seller is queried only after the buyer accepts the trade, effectively minimizing the amount of information that the intermediary has to collect from the seller while still achieving an optimal regret rate.

**Questions:**

I understand that the focus here is on minimizing feedback from the seller for obvious practical reasons (lines 52-61). However, it seems to me that the roles of the buyer and the seller are theoretically interchangeable in your setup. Therefore, I'm left with the question of whether, theoretically, your algorithm can be easily adopted in a scenario where we want to minimize the feedback from the buyer (i.e., query the buyer only after the seller accepts the trade). In this case, I recommend briefly mentioning it somewhere in the paper, dropping "Don't Waste Seller's Time" from the title, and keeping only "Online Bilateral Trade With Minimal Feedback" for clarity.

**Ethical Concerns:**

["NO or VERY MINOR ethics concerns only"]

**Final Justification:**

The authors confirmed that the roles of sellers and buyers are interchangeable, which adds value to the proposed framework. Moreover, the authors provided numerical experiments that furhter strenghten their work. The only reason why I maintain my original evaluation of 5, instead of increasing to 6, is that I see the contribution focused on a specific sub-area instead of one or more areas of AI.

**Limitations:**

Yes

**Quality:**

4

**Strengths And Weaknesses:**

The problem formulation is precise. The paper is well-written, and it has been a pleasure to read. Although the work is strongly based Cesa-Bianchi et al. [2024a] and incremental, the extension to asynchronous interaction is new to the best of my knowledge. The result of achieving an optimal regret rate with minimal feedback is strong.

---

> ### Author Rebuttal · Authors · 2025-07-30
>
> **Re: "I understand that the focus here is on minimizing feedback from the seller for obvious practical reasons (lines 52-61). However, it seems to me that the roles of the buyer and the seller are theoretically interchangeable in your setup. Therefore, I'm left with the question of whether, theoretically, your algorithm can be easily adopted in a scenario where we want to minimize the feedback from the buyer ..."**
>
> We agree with the reviewer, the roles of sellers and buyers are interchangeable, we will specify this better.

---

> > ### Comment · Reviewer_vJmF · 2025-08-01
> >
> > I thank the authors for their confirmation, and I confirm my evaluation of the paper.

---

### Official Review · Reviewer_VQjt · 2025-06-25

**Clarity:** 3
**Significance:** 2
**Originality:** 2
**Rating:** 3
**Confidence:** 3

**Summary:**

The authors propose an asynchronous protocol for the online bilateral trade problem, designed to be more efficient by not wasting a seller's time. The proposed asynchronous mechanism queries the buyer and only approaches the seller if the buyer agrees to the price. The main contribution is an algorithm for this new setting that achieves a regret that matches the known lower bound.

**Questions:**

See weakness.

**Ethical Concerns:**

["NO or VERY MINOR ethics concerns only"]

**Final Justification:**

After reading the response, my comment and scores remain the same.

**Limitations:**

Yes.

**Paper Formatting Concerns:**

NA.

**Quality:**

2

**Strengths And Weaknesses:**

Strength:
1. The paper's primary strength is the introduction of the "Don't Waste Seller's Time" asynchronous protocol. This is a more efficient model.

2. The analysis presents a technical contribution in handling the "censored" seller feedback. The authors overcome the challenge of having potentially fewer samples for the seller's valuation by showing that when this occurs, the buyer's acceptance probability is also low.

Weakness:
1. I don't quite understand how the examples mentioned in the Introduction—such as online freelance marketplaces (Upwork, Fiverr), ride-sharing platforms (Uber, Lyft, Grab), and rental intermediaries (Airbnb)—correspond to the framework presented in this paper. Could you explain in more detail how the paper's algorithm would be applied in these domains? Based on my own user experience, these platforms appear to display the seller's price directly, which seems to be a complete mismatch with the framework presented in this paper, where a broker sets a single price for both parties.

2. The technical novelty is not clear, as the framework and algorithmic approach of this paper heavily overlap with Section 5 of 'Bilateral Trade: A Regret Minimization Perspective' by Cesa-Bianchi et al. [2024a]. Even thought the authors argue that the samples from sellers are asynchronous and asymmetric, the core concepts are the same: the decomposition principle, the grid-based approach, and the strategy of first estimating the integral terms and then using a bandit-style method to explore and exploit the local terms. The amount of new content proposed in this paper is quite limited.

3. This paper has no experiments at all, neither on synthetic nor on real-world data. Therefore, without experiments, it's unclear how the algorithm actually performs in practice or how it behaves with a limited amount of data.

---

> ### Author Rebuttal · Authors · 2025-07-30
>
> **Re: "I don't quite understand how the examples mentioned in the Introduction—such as online freelance marketplaces (Upwork, Fiverr), ride-sharing platforms (Uber, Lyft, Grab), and rental intermediaries (Airbnb)—correspond to the framework presented in this paper. Could you explain in more detail how the paper's algorithm would be applied in these domains? Based on my own user experience, these platforms appear to display the seller's price directly, which seems to be a complete mismatch with the framework presented in this paper, where a broker sets a single price for both parties. "**
>
> We are not claiming that Upwork/Fiverr, Uber/Lyft/Grab, or Airbnb currently run our mechanism (a single price posted symmetrically to both sides). Many use two distinct pricing/selection steps and monetize the spread between buyer and seller prices. We propose a viable alternative: the platform quotes the buyer a transfer price $p$ and, only if the buyer accepts, pings a seller at the same $p$, yielding exactly the feedback we analyze (buyer bit each round; censored seller bit only after buyer acceptance). In this zero‑spread design, we explicitly maximize gain from trade (the traders’ joint surplus under strong budget balance), so the platform can credibly commit to optimizing user welfare rather than extracting a spread, a stance that plausibly increases entry and retention of traders. Crucially, monetization is decoupled from $p$: the platform can earn via advertising or sponsored placement, subscriptions/memberships, or fixed per‑trade fees independent of $p$, all without changing the acceptance logic or our theoretical guarantees.
>
> **Re: "The technical novelty is not clear, as the framework and algorithmic approach of this paper heavily overlap with Section 5 of 'Bilateral Trade: A Regret Minimization Perspective' by Cesa-Bianchi et al. [2024a]. Even thought the authors argue that the samples from sellers are asynchronous and asymmetric, the core concepts are the same: the decomposition principle, the grid-based approach, and the strategy of first estimating the integral terms and then using a bandit-style method to explore and exploit the local terms. The amount of new content proposed in this paper is quite limited."**
>
> We agree with the reviewer that our algorithm shares similarities with the one by Cesa‑Bianchi et al. 2024a, as also noted in our paper. However, the main contribution of our work is analytical: we establish regret bounds comparable to those of Cesa‑Bianchi et al. under strictly weaker, censored (1.5) feedback. We remark that under this 1.5 feedback the algorithm they propose (Scouting Bandits) does not even run. Additionally, the 1.5‑feedback setting poses unique challenges: seller feedback at a given price $p$ is observed only when the buyer accepts. Consequently, some prices receive very few seller‑side samples, inflating uncertainty about willingness to sell and potentially hindering optimal rates. To address this, we derive (via multiplicative Chernoff bounds) a lower bound on the number of rounds in which seller feedback is observed at price $p$. We stress that because the underlying variables are Bernoulli with potentially small means, additive Hoeffding‑type bounds (as employed by Cesa‑Bianchi et al., 2024a) are ineffective in this rare‑event regime. Together with the fact that when seller‑side confidence intervals are large the buyer‑acceptance probability is low, a careful analysis shows that these two competing effects in fact compensate each other, allowing us to recover the same regret rates despite censoring.
>
>
> **Re: "This paper has no experiments at all, neither on synthetic nor on real-world data. Therefore, without experiments, it's unclear how the algorithm actually performs in practice or how it behaves with a limited amount of data."**
>
> In the following table, we report the numerical values obtained by running our algorithm (using 1.5-bit feedback) and the state-of-the-art algorithm by Cesa-Bianchi et al. (2024), which uses 2-bit feedback. In each instance where we ran the two algorithms, the buyer and seller valuations were drawn from Beta distributions with the parameters $\alpha_b, \alpha_s, \beta_b, \beta_s$ specified below. Each experiment was repeated $n = 5$ times, and the final regret values correspond to the mean over these $n = 5$ runs.
>
> | Parameter                                        | Instance 1              | Instance 2               | Instance 3                          | Instance 4                          | Instance 5                          | Instance 6                          |
> |--------------------------------------------------|--------------------------|---------------------------|--------------------------------------|--------------------------------------|--------------------------------------|--------------------------------------|
> | Time horizon                                     | 10000                   | 50000                    | 10000                               | 50000                               | 10000                               | 50000                               |
> | ($\alpha_s$, $\beta_s$)                          | (5.0, 10.0)              | (5.0, 10.0)               | (10.0, 10.0)                         | (10.0, 10.0)                         | (2.0, 3.0)                           | (2.0, 3.0)                           |
> | ($\alpha_b$, $\beta_b$)                          | (15.0, 10.0)             | (15.0, 10.0)              | (15.0, 10.0)                         | (15.0, 10.0)                         | (10.0, 10.0)                         | (10.0, 10.0)                         |
> | **Cumulative Regret ± std (ours, 1.5 feedback)** | $199.56 \pm 17.14$       | $714.44 \pm 73.07$        | $135.21 \pm 22.12$                   | $326.51 \pm 83.98$                   | $168.53 \pm 28.78$                   | $439.17 \pm 136.17$                 |
> | Cumulative Regret ± std (2-bit feedback)         | $732.00 \pm 21.41$       | $2253.76 \pm 75.22$       | $548.83 \pm 16.35$                   | $2380.95 \pm 98.14$                  | $583.48 \pm 27.58$                   | $2502.37 \pm 52.19$                 |
>
> We observe that the regret incurred by our algorithm is lower than that of Cesa-Bianchi et al. 2024a. While this may appear counterintuitive (since we use less feedback than the Scouting Bandits algorithm of Cesa-Bianchi et al.) the improvement stems from a key difference: after the initial exploration phase, we eliminate arms that are guaranteed to be suboptimal. This pruning step, absent in Scouting Bandits, allows us to restrict the subsequent bandit phase to the reduced set $K^\diamond$ (see Line 12), which can be significantly smaller than the original set of arms. In contrast, Cesa-Bianchi et al. 2024a run their algorithm over the full set, whose size is $T^{1/3}$.

---

> > ### Comment · Reviewer_VQjt · 2025-08-06
> > **Thanks for the response**
> >
> > I thank the authors for the response, which provides a better clarification of the motivation. But my concern on technical novelty still remains.

---

### Official Review · Reviewer_KuvA · 2025-07-02

**Clarity:** 4
**Significance:** 3
**Originality:** 3
**Rating:** 5
**Confidence:** 3

**Summary:**

The paper studies the problem of online bilateral trading in an asynchronous setting. There are three agents involved; the seller, the buyer and the broker. At each round $t$, the broker sets a price $p_t$ and the trade occurs if $S_t<p_t<B_t$. The problem has been studied under different feedback models, namely the full feedback, where the broker observes both buyer’s and seller’s valuation, the 2-bit model where the broker observes separately the boolean random variable $X_1 = p<B$ and $X_2= p>S$ and the 1-bit model where the broker only observes the ternary boolean $X = S<p<B$. They consider an asynchronous model in between 1-bit and 2-bit, where the broker only observes the boolean $X_2= p>S$ if the buyer first accepts the price $p_t$.
The authors give an online learning algorithm that operates over a uniform grid of size $K=\lceil T^{1/3} \rceil $.They prove that their online bilateral trading algorithm in the latter feedback setting that achieves regret of $O(T^{2/3})$.

**Questions:**

Have you considered studying information-accuracy tradeoffs such as:
- How much mutual information about the optimal price does each feedback bit supply ?
- What regret rates are achievable if you deliberately limit feedback to, say, one bit every other round, or to noisy bits?
- How much “value” does each bit contribute toward narrowing down the optimal price?

Framing the problem in an information-theoretic bandit lens, that is, bounding regret in terms of cumulative mutual information between observations and the optimal actio, would let you precisely quantify when and why extra seller signals cease to help.

**Ethical Concerns:**

["NO or VERY MINOR ethics concerns only"]

**Final Justification:**

The authors fully addressed my questions. I remain very positive for the novelty of this paper.

**Limitations:**

yes

**Paper Formatting Concerns:**

There are no formatting issues

**Quality:**

3

**Strengths And Weaknesses:**

### Strengths
- The paper is well written and well organized.
- The idea of exploring a feedback model between 1-bit and 2-bits is novel in such a setting.

### Weaknesses
- It would be valuable to discuss how sensitive the algorithm is to violations (e.g., if densities have unbounded spikes) and whether weaker smoothness conditions (e.g., Hölder) suffice.

---

> ### Author Rebuttal · Authors · 2025-07-30
>
> **Re: "It would be valuable to discuss how sensitive the algorithm is to violations (e.g., if densities have unbounded spikes) and whether weaker smoothness conditions (e.g., Hölder) suffice."**
>
> Great question. Without regularity, sublinear regret cannot be guaranteed uniformly. In particular, if one allows unbounded density spikes, we can tune the spike height with the time horizon to force linear regret for any learning algorithm. Formally, we can adapt the lower‑bound idea of Cesa‑Bianchi et al. (2024a, Thm. 6) to smoothed distributions: let the seller pdf be:
>
> $f_S(u)= M  \mathbf{1}_{[0,\frac{1}{2M}]}(u) + M  \mathbf{1} _{[x-\frac{1}{2M}, x]} (u)$,
>
> and the buyer pdf
>
> $f_B(u)=M   \mathbf{1}_{[1-\frac{1}{2M}\,1]}(u) +M \mathbf{1} _{[x\,x+\frac{1}{2M}]}(u)$,
>
> so both integrate to 1, for some $x$ near $\frac{1}{2}$. First, by Yao’s principle, it suffices to consider deterministic algorithms in this i.i.d. setting. If we choose $M=2^{\Theta(T)}$, then for any deterministic algorithm there exists an $x$ such that the algorithm never plays in the optimal interval $[x-\tfrac{1}{2M},x+\tfrac{1}{2M}]$, yielding $\Omega(T)$ regret. Intuitively, locating that interval requires $\Omega(\log M)=\Omega(T)$ informative plays (a “binary‑search‑like” effort); until then, the per‑round regret remains bounded away from zero. Equivalently: no uniform sublinear rate is possible over classes that permit arbitrarily tall, narrow spikes.
> On the other hand, weaker smoothness suffices. If the cdfs $F_S$ and $F_B$ are $\alpha$-Hölder with constant $L$, one can still control the discretization error. Using Lemma 1 of Cesa‑Bianchi et al. (2024a), for $0\le p<q\le 1$,
>
>
> $$|g(q) - g(p)|  =\left| F_S(p) \int_p^1 (1 - F_B(u))du + (1 - F_B(p)) \int_0^p F_S(u)du - \left( F_S(q) \int_q^1 (1 - F_B(u)) du + (1 - F_B(q)) \int_0^q F_S(u) du \right) \right|$$
> $$\le
> \max \left( (F_B(q) - F_B(p)) \int_0^q F_S(u) du, (F_S(q) - F_S(p)) \int_q^1 (1 - F_B(u)) du \right) + \max \left( F_S(p) \int_p^q (1 - F_B(u)) du, (1 - F_B(p)) \int_p^q F_S(u) du \right)
> $$
> $$
> \le L |q - p|^\alpha + |q - p| \le (L + 1) |q - p|^\alpha.
> $$
>
>
> where the last inequality uses $|q-p|\le1$ and $\alpha\in(0,1]$. Thus $g$ is $\alpha$-Hölder, the discretization bias scales as $K^{-\alpha}$, and with appropriate tuning of $K$ one obtains sublinear regret.
> Whether these rates are minimax‑optimal under Hölder cdfs is an interesting open question we leave for future work.
>
>
>
>
> **"Re: Have you considered studying information-accuracy tradeoffs such as: How much mutual information about the optimal price does each feedback bit supply ? What regret rates are achievable if you deliberately limit feedback to, say, one bit every other round, or to noisy bits? How much “value” does each bit contribute toward narrowing down the optimal price?"**
>
> Very interesting questions! Unfortunately, we are not familiar with the literature analyzing online learning algorithms through the information-theoretic lens; thus, we cannot fully answer the questions raised by the Reviewer. However, we believe that such questions are very relevant to our paper, as they go towards the direction of deeply understanding what is the contribution of asymmetric feedback to learning the optimal price. We will surely investigate these aspects in future works.

---

### Official Review · Reviewer_WMxL · 2025-07-02

**Clarity:** 4
**Significance:** 3
**Originality:** 3
**Rating:** 5
**Confidence:** 3

**Summary:**

This paper studies the online bilateral trading problem, where a market maker (“broker”) aims at facilitating trades between pairs of buyers and sellers (drawn i.i.d.) arriving over time $t=1,2,\ldots, T$ with a price $p_t$ in the middle. The performance is measured by regret on the gain from trade against the best fixed price in hindsight. The authors introduced an asynchronous feedback protocol, where the broker always observes the buyer’s willing-to-trade $1[p_t\leq B_t]$ but only observes a  feedback from the seller’s side $1[S_t\leq p_t]$ if the buyer has accepted the price, thus formulating a 1.5-bit-feedback censored feedback model. In this paper, the authors propose an algorithm that achieves $\tilde{O}(T^{2/3})$ regret, which is near-optimal even for the two-bit feedback problem setting. The algorithm applies a pure-exploration-then-UCB strategy.

**Questions:**

### Questions:

Please see “weakness” above. Also, what is the computational complexity of the algorithm, particularly the $K$-dependence as you adopt the grid-based approach?

### Suggestions:

Some typos to fix: e.g. Line 138 (the equation): “sellers’s”.

**Ethical Concerns:**

["NO or VERY MINOR ethics concerns only"]

**Final Justification:**

The rebuttal resolves my concerns and I lift my grading.

**Limitations:**

yes

**Quality:**

3

**Strengths And Weaknesses:**

### Strength:

This problem is novel and well-motivated as it is only necessary to ask the seller’s side when the buyer has approved the trading. The 1.5-bit feedback model (as sometimes 1 and sometimes 2) is more difficult than the previous two-bit feedback model, and it is non-trivial to achieve the same regret rate. Regarding the identical regret rate (in big O notation), it also justifies that the 1.5-bit model is not losing too much information from the two-bit model, which aligns with many existing literatures on pricing and trading with censored feedback. Besides, the algorithmic analysis is rigorous and sound, with assumptions, theorems and lemmas clearly stated. The paper is overall well written.

### Weakness:

I think the most concern raised from my side is the lack of numerical experiments. Although this work is theoretic-oriented and the theoretical results are contributive, I still think it is particularly necessary for this paper to have at least simulations. The reason is simple as I would like to know whether the feedback censoring effect on the seller’s side is harming the reward or affecting the regret in practice (although the regret rate is the same in big O notation). There are two possible results as I expect: (1) The cumulative reward of proposed algorithm decreases faster than the baseline oracle, leading to a larger regret on the 1.5-bit problem than the original two-bit problem. This means that the algorithm suffers more than the baseline oracle from the censoring effect. (2) The regret is even smaller than the two-bit case, indicating that the baseline oracle suffers more than the algorithm (or the player in general). While (1) could be more reasonable, I wonder if (2) indeed happens, then it might still not be a “Waste of Seller’s Time” to ask both sides simultaneously. Do you agree with that?

Besides, there are some drawbacks from the modeling assumptions, e.g. the finite price set, the i.i.d. assumption, the bounded density, etc. Some of them are not well justified.

---

> ### Author Rebuttal · Authors · 2025-07-30
>
> **Re: "I think the most concern raised from my side is the lack of numerical experiments. Although this work is theoretic-oriented and the theoretical results are contributive, I still think it is particularly necessary for this paper to have at least simulations. "**
>
> In the following table, we report the numerical values obtained by running our algorithm (using 1.5-bit feedback) and the state-of-the-art algorithm by Cesa-Bianchi et al. (2024), which uses 2-bit feedback. In each instance where we ran the two algorithms, the buyer and seller valuations were drawn from Beta distributions with the parameters $\alpha_b, \alpha_s, \beta_b, \beta_s$ specified below. Each experiment was repeated $n = 5$ times, and the final regret values correspond to the mean over these $n = 5$ runs.
>
> | Parameter                                        | Instance 1              | Instance 2               | Instance 3                          | Instance 4                          | Instance 5                          | Instance 6                          |
> |--------------------------------------------------|--------------------------|---------------------------|--------------------------------------|--------------------------------------|--------------------------------------|--------------------------------------|
> | Time horizon                                     | 10000                   | 50000                    | 10000                               | 50000                               | 10000                               | 50000                               |
> | ($\alpha_s$, $\beta_s$)                          | (5.0, 10.0)              | (5.0, 10.0)               | (10.0, 10.0)                         | (10.0, 10.0)                         | (2.0, 3.0)                           | (2.0, 3.0)                           |
> | ($\alpha_b$, $\beta_b$)                          | (15.0, 10.0)             | (15.0, 10.0)              | (15.0, 10.0)                         | (15.0, 10.0)                         | (10.0, 10.0)                         | (10.0, 10.0)                         |
> | **Cumulative Regret ± std (ours, 1.5 feedback)** | $199.56 \pm 17.14$       | $714.44 \pm 73.07$        | $135.21 \pm 22.12$                   | $326.51 \pm 83.98$                   | $168.53 \pm 28.78$                   | $439.17 \pm 136.17$                 |
> | Cumulative Regret ± std (2-bit feedback)         | $732.00 \pm 21.41$       | $2253.76 \pm 75.22$       | $548.83 \pm 16.35$                   | $2380.95 \pm 98.14$                  | $583.48 \pm 27.58$                   | $2502.37 \pm 52.19$                 |
>
> We observe that the regret incurred by our algorithm is lower than that of Cesa-Bianchi et al. 2024a. While this may appear counterintuitive (since we use less feedback than the Scouting Bandits algorithm of Cesa-Bianchi et al.) the improvement stems from a key difference: after the initial exploration phase, we eliminate arms that are guaranteed to be suboptimal. This pruning step, absent in Scouting Bandits, allows us to restrict the subsequent bandit phase to the reduced set $K^\diamond$ (see Line 12), which can be significantly smaller than the original set of arms. In contrast, Cesa-Bianchi et al. 2024a run their algorithm over the full set, whose size is $T^{1/3}$.
>
> **Re: "Besides, there are some drawbacks from the modeling assumptions, e.g. the finite price set, the i.i.d. assumption, the bounded density, etc. Some of them are not well justified."**
>
> From a mathematical point of view, rescaling prices to lie within $[0,1]$ is a standard assumption in the bilateral trade literature, adopted primarily for convenience. We note, however, that our algorithm can be readily adapted to any bounded interval $[0,M]$ through an appropriate rescaling. If valuations and prices instead range over $[0, +\infty)$, and no further assumptions are made, one can easily show that any algorithm must incur linear regret. That said, from an economic point of view, it is arguably realistic to assume the existence of an upper bound $M$ on the value of any given good.
>
> From a mathematical point of view, removing the i.i.d. assumption is known to lead to linear regret rates (see Theorem 5 in Cesa-Bianchi et al. 2024a). That said, from an economic point of view, the i.i.d. assumption can be interpreted as modeling interactions between a broker and a large, stable market. Specifically, if the population is sufficiently large that each seller–buyer pair can be viewed as independently drawn (i.e., sampling with replacement), and if the market is stable over time so that the distribution of valuations remains unchanged, then the overall process can reasonably be treated as i.i.d..
>
> Again, from a mathematical point of view, removing the bounded density assumption (Lipschitzness of the cdf) is known to lead to linear regret rates (see Theorem 6 in Cesa-Bianchi et al. 2024a). Economically, this assumption rules out highly concentrated valuation distributions and instead reflects the idea that agents are sufficiently heterogeneous in how they value the good. This heterogeneity is a natural feature of many real-world markets, where valuations emerge from the interaction of multiple factors, such as individual preferences, time constraints, opportunity costs, available alternatives, expectations, budget limitations, and even random noise. For instance, two sellers offering the same service on platforms like Airbnb or Fiverr may have different reservation prices due to variations in location, experience, urgency, or subjective valuation of their time. Likewise, buyers differ in urgency, purpose, willingness to pay, and familiarity with the platform. Moreover, digital marketplaces are typically designed to support broad participation, encouraging diversity in both supply and demand. The bounded density assumption essentially captures markets containing a wide, continuous spectrum of agent types.
>
> We will add this discussion about the implications of our assumptions in the revised version.
>
>
> **Re: "What is the computational complexity of the algorithm, particularly the dependence as you adopt the grid-based approach?"**
>
> The per-round computational complexity of our algorithm is linear $K=T^{1/3}$, with $K$ being the size of the grid employed by our algorithm.
>
>
>
> Nicolò Cesa-Bianchi, Tommaso Cesari, Roberto Colomboni, Federico Fusco, and Stefano Leonardi. Bilateral trade: A regret minimization perspective. Mathematics of Operations Research, 49(1): 171–203, 2024a.

---

> ### Comment · Reviewer_WMxL · 2025-08-04
>
> That sounds really excited (and counterintuitive which is not bad). I am supporting this work with an increased score, while I also have the following suggestions:
>
> (1) For the numerical experiments, maybe it is possible to also improve the baseline (Cesa-Bianchi et al. 2024) with your policy elimination design. This helps explain whether the indeed *optimal* regret (from the perspective of problem hardness) of the 1.5-bit feedback problem is smaller than that of the 2-bit one.
>
> (2) Please not only print the regret comparisons but also the **reward** comparisons on two aspects: one for baselines, and the other for algorithms. This helps indicate how the 1.5-bit feedback contrasts the 2-bit one.

---

### Decision · Program_Chairs · 2025-09-17

**Decision:**

Accept (poster)

**Comment:**

Reviewers are positive about paper's writing and like the results. However, there are concerns on paper's technical contributions and practical relevance. Overall, the paper is slightly above borderline.

I share reviewers' concerns. And here are some of my suggestions for improving practical relevance (not required for a theory-focused paper, but good to have):
1) Offer extra evidence that the online bilateral trade algorithms with more or less feedback are used in any practical scenarios, and how sellers react to them. It's impossible to judge if "Don’t Waste Seller’s Time" is an important/relevant issue to solve to start with, from a practical perspective.
2) It would be helpful to have the numerical simulation in rebuttal together with more baselines, including strategies that care less about asymptotic performance but more on performance with reasonable size inputs.